

# Population dynamics of Schrödinger cats

Foster Thompson[1][⋆] and Alex Kamenev[1,2][†]

**1** School of Physics and Astronomy, University of Minnesota, Minneapolis, MN 55455, USA
**2** William I. Fine Theoretical Physics Institute, University of Minnesota,
Minneapolis, MN 55414, USA

⋆ thom7385@umn.edu , † kamen002@umn.edu

## Abstract

We demonstrate an exact equivalence between classical population dynamics and Lindbladian evolution admitting a dark state and obeying a set of certain local symmetries. We then introduce *quantum population dynamics* as models in which this local symmetry condition is relaxed. This allows for non-classical processes in which animals behave like Schrödinger's cat and enter superpositions of live and dead states, thus resulting in coherent superpositions of different population numbers. We develop a field theory treatment of quantum population models as a synthesis of Keldysh and third quantization techniques and draw comparisons to the stochastic Doi-Peliti field theory description of classical population models. We apply this formalism to study a prototypical "Schrödigner cat" population model on a $d$-dimensional lattice, which exhibits a phase transition between a dark extinct phase and an active phase that supports a stable quantum population. Using a perturbative renormalization group approach, we find a critical scaling of the Schrödinger cat population distinct from that observed in both classical population dynamics and usual quantum phase transitions.

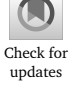

# 1 Introduction

Recent years are marked by the accelerated pace of building quantum computation platforms with an ever increasing number of interconnected qubits [1–11]. A further development of such platforms and their utilization for implementation of practical quantum algorithms require a better understanding of the *many-body* collective dynamics of these devices. Unlike textbook spins, or other quantum particles, the qubits are often macroscopic objects with huge number of internal degrees of freedom. The inescapable consequence of this fact is that the qubits' density matrix does *not* evolve in a pure unitary way, described by the von Neumann equation. Indeed, tracing out numerous degrees of freedom, which are not explicitly controlled by the device (aka bath), results in a *reduced* density matrix for the connected qubit assembly. In the simplest approximation, which disregards non-Markovian memory effects of the bath, the reduced density matrix evolves according to the Lindblad equation [12–14]. The latter incorporates the von Neumann Hamiltonian part as well as bath-induced dissipative terms, expressed in terms of the so-called quantum jump operators, $\hat{L}_\nu$.

Despite a lot of recent developments [15–17], the many-body theory of the Lindbladian evolution is still in its infancy. In particular, we are still lacking a basic intuition and even a proper vocabulary to discuss dynamical phases and corresponding phase transitions in such models. Within this framework one particular regime of interest is many-body Lindbladians that protect certain pure states, known in this context as dark states, from all fluctuations [10]. Such states are the analogues of classical absorbing states [18], which are totally dynamically inert and, once entered, cannot be left. The existence of a dark state of a particular dissipative quantum dynamics does not guarantee that such a state is actually reached at long times for generic initial conditions. This occurs when the particular combination of jump operators and Hamiltonian drives the system away from the dark state, rendering it dynamically inaccessible. Regimes with accessible and inaccessible dark states are separated by non-equilibrium phase transitions, a feature shared with models of classical absorbing states [16].

Quantum dark states and their stability as well as absorbing state phase transitions in a quantum setting have been a subject of several recent publications. In particular, there have been theoretical studies of first order [19] and continuous [20,21] dark state phase transitions in Lindbladian spin systems, as well as a variety of numerical studies [22–24]. Various bosonic Lindbladians have been considered as analogues of classical reaction models, and have been shown to exhibit absorbing state transitions and non-classical relaxation dynamics [25–28].

Fermionic models were examined in Ref. [29] and subsequently in Refs. [30,31], where it was argued that certain classes of fermionic models generically display unstable dark states. An additional motivation for the study of such models is that they can be interpreted as dissipative quantum protocols for steering towards and stabilizing certain desirable quantum states. In this context, a dark state transition represents a limit beyond which the particular protocol under consideration breaks down and the desired target state is no longer obtainable. This is of interest as a potential mean of robustly storing quantum information [32–36], or realizing a quantum order in open systems [37–39]. We also mention related works applying the machinery of classical population dynamics and absorbing state transitions to study properties of quantum circuits [40–43].

Many of the essential features of quantum theories with dark states are also present in classical population models. Classical population dynamics is a stochastic theory, formulated in terms of the probability distributions of certain population outcomes. In such models, the state with zero animals is an absorbing state. The dynamical stability of this state is of central interest and determines whether or not the population goes extinct at long times. Dynamics of the classical probability distributions are governed by the so-called Master equation. The latter is essentially a kinetic equation, which describes rates of "in" and "out" events changing probability of a given outcome. It was realized long ago that one can conveniently represent such a Master equation as an imaginary time Schrödinger evolution with a non-Hermitian effective Hamiltonian. In the earliest incarnation, known as Doi-Peliti representation [44,45], such a Hamiltonian is expressed through the canonical $\hat{a}$ and $\hat{a}^\dagger$ operators which decrease and increase the population size by one animal, correspondingly. This technique proved to be extremely useful to develop field theoretical techniques to describe spatially extended population dynamics models and classify their dynamical non-equilibrium phase transitions [18,46–50].

In this work we advocate for a common language of "*quantum population dynamics*" which serves as a convenient tool to visualise and describe a multitude of phenomena observed in quantum dissipate dynamics with dark states. We show that there is an equivalent way of representing classical population models as dynamics of diagonal elements of a fictitious quantum density matrix. The latter evolves according to a certain Lindblad equation. As explained below, if all quantum jump operators, $\hat{L}_\nu$, respect a certain local symmetry, dynamics of the diagonal elements decouples from the rest of the density matrix. Moreover, evolution of these diagonal elements is described by a Master equation for some set of the population reaction rules. There is thus a one to one correspondence between classical population dynamics rules and the Lindbladian evolution with a proper set of the symmetry-respecting quantum jump operators.

The advantage of this representation is that it may be naturally generalized to the *quantum* population dynamics. The latter is formulated in terms of dynamical rules which allow for populations of "Schrödinger cats", described by coherent superpositions of populations with different sizes (e.g., a superposition of a dead cat, $|0\rangle$, and a live cat, $|1\rangle = \hat{a}^\dagger |0\rangle$). To achieve such non-classical population outcomes, one needs to allow for the quantum jump operators which violate the local symmetry, mentioned above. Such a vantage point views any (bosonic) many-body Lindbladian (supporting a dark state) as an equivalent set of quantum population reaction rules.

This point of view implies that the "quantumness," being a symmetry breaking phenomenon, may be a *relevant* perturbation of the classical dynamics. This opens the possibility that hybrid quantum-classical models may exhibit universality classes which are distinct from that of the pure classical population models. On the other side, pure quantum models (such as, e.g., quantum magnets, or Bose condensates) are often unstable against dissipative Lindbladian terms. In many such cases, the addition of incoherent processes drives quantum models towards known classical equilibrium or non-equilibrium dynamic universality classes [51,52]

associated with stochastic processes (several notable examples are provided by [53–55], for a more exhaustive review see [16]). This is however not always the case, and in some cases the combination of coherent and dissipative processes in Lindbladian dynamics may exhibit phase transitions with universal behavior distinct from both pure classical and pure quantum models, for example [53,56–58]. As we argue below, a generic bosonic Lindbladian dynamics with a dark state (i.e. Schrödinger cats population dynamics) provides an example of novel critical behavior, distinct from that observed in both classical and closed quantum systems.

To this end, we develop a field-theoretic approach, similar in form to the Doi-Peliti formalism for the classical stochastic systems, for the treatment of quantum dissipative dynamics with dark states. Our approach synthesizes quantum many-body Keldysh techniques [52] with the third quantization description of the Lindbladian evolution [59–64]. It allows us to identify a distinct "Schrödinger cat" universality class, characteristic for models without any extra symmetries or dynamical constraints, beyond the existence of the extinct dark state. We derive a field theory for such models in $d$ spatial dimensions and develop a renormalization group (RG) treatment for the phase transition between the dead and live phases. In yet other words, this is the phase transition between a trivial dark state (a classical absorbing state) and a nontrivial, dynamically stable, essentially quantum state of the many-body system. The result of this analysis is a set of critical indexes, obtained in the $\epsilon = 4 - d$ expansion, which is distinct from both pure classical (i.e. directed percolation) and quantum models.

The structure of the manuscript is as follows. In Section 2 we establish the correspondence between classical population dynamics and quantum Lindbladians with *weak* local $U(1)$ symmetries. We then introduce quantum population dynamics as a generalization of these classical models and discuss various examples of quantum population processes. Section 3 discusses the structure of the field theories for models with absorbing states. We begin with a review of the Doi-Peliti field theory description of the classical directed percolation universality class and then develop a complimentary treatment of a quantum bosonic theory with dark states, emphasizing connections between the two formalisms. This machinery is subsequently applied in 4 to derive and study a universal effective field theory for the Scrhödinger cat population dynamics of a single species. We identify a dark state phase transition in this theory and derive its critical exponents, which turn out to be distinct from those of the classical population dynamics.

## 2   Reaction models and Lindbladian dynamics

Classical population dynamics describes the behaviour of animal species living in their environment, in which they can reproduce, disperse, compete, die, etc. Formally, reaction models are a class of stochastic processes in which different agents incoherently evolve according to a set of microscopic reaction rules. A population dynamics is a specific type of such reaction models that possesses an *absorbing* state with zero population for all species.

In this section we establish a correspondence between certain types of Lindbladians and classical reaction models. The main idea may be summarized as follows: consider a Lindbladian with a *weak* $U(1)$ symmetry (or a multiple copies of such a symmetry - e.g., one for each species). If all jump operators posses a definite charge under at least one copy of $U(1)$, then the dynamics of states diagonal in the charge basis is equivalent to a classical reaction model. If, in addition, the zero charge vacuum is a dark state of such a Lindbladian, the latter is equivalent to a classical population dynamics.

Note that many reaction models put limits on the total number of each agent. Such constrained reaction models are often relevant in the quantum context in which the local degrees of freedom are qubits, which in the present context is equivalent to an agent which can have a local population of only zero or one. Here we work with reaction models in which the populations of each agent is unconstrained both for conceptual clarity and for the technical advantage of being describable using bosonic field theory methods. Nevertheless, the main lessons of this section hold true also for constrained reaction models and population dynamics. All arguments presented can be readily adapted to include constrains by replacing the canonical bosonic operators with appropriate finite dimensional qudit analogues.

## 2.1 Classical reaction models and absorbing states

Here we briefly review basic formal details of classical reaction models. Consider a set of $N$ different types of agents or species denoted as $A_j$ with $j = 1, \dots, N$. Groups of agents may interact according to reaction rules,

$$\sum_j^N m_j A_j \xrightarrow{\gamma} \sum_j^N m'_j A_j \,, \tag{1}$$

which signifies a reaction that occurs at rate $\gamma$ in which $m_1$ of agent $A_1$, $m_2$ of agent $A_2$, etc. react to produce $m'_1$ of agent $A_1$, $m'_2$ of agent $A_2$, etc. The null symbol $\emptyset$ will be used for rules in which there are either no reactants or products.

The space of states of classical reaction models is formally equivalent to a bosonic Fock space, with a single bosonic mode for each agent. A state vector $|\mathbf{n}\rangle = |n_1, n_2, \dots\rangle$ denotes the state with $n_j$ of agent $A_j$. Linear combinations of different number states represent *statistical* superpositions of different number states. The state of a system is a classical probability distribution $\mathcal{P}$, which is a represented by a general ket $|\mathcal{P}\rangle = \sum_{\mathbf{n}} \mathcal{P}_{\mathbf{n}} |\mathbf{n}\rangle$ which satisfies $\sum_{\mathbf{n}} \mathcal{P}_{\mathbf{n}} = 1$. In a slight abuse of notation, this normalization condition is conveniently represented as an overlap with the "state" with $\mathcal{P}_{\mathbf{n}} = 1$, so that $\langle 1 | \mathcal{P} \rangle = 1$.

Following the standard operator formalism, introduced originally in [65], one may define a number operator $\hat{n}_j$ for each agent, which has the number states as eigenvectors:

$$\hat{n}_j |\mathbf{n}\rangle = n_j |\mathbf{n}\rangle \,. \tag{2}$$

Each $\hat{n}_j$ has a corresponding canonically conjugated operator $\hat{p}_j$, the exponential of which defines the raising (or lowering) operator,

$$\mathrm{e}^{\pm \hat{p}_j} |\mathbf{n}\rangle = |\dots, n_j \pm 1, \dots\rangle \,. \tag{3}$$

The two satisfy the standard bosonic commutation rule $[\hat{n}_j, \hat{p}_i] = \delta_{ij}$.

Dynamics set by the reaction rules are represented by a classical Master equation:

$$\partial_t |\mathcal{P}\rangle = \hat{\mathcal{W}} |\mathcal{P}\rangle \,. \tag{4}$$

The (non-Hermitian) evolution generating operator $\hat{\mathcal{W}}$ is a sum of operators that individually encode each reaction rule $\hat{\mathcal{W}} = \sum_{\nu} \gamma_{\nu} \hat{\mathcal{W}}_{\nu}$. Typically, one chooses for a general reaction rule as in Eq. (1):

$$\hat{\mathcal{W}}_{\nu} = \left( \mathrm{e}^{-\sum_j (m_{\nu j} - m'_{\nu j}) \hat{p}_j} - 1 \right) \prod_j \frac{\hat{n}_j!}{m_{\nu j}! (\hat{n}_j - m_{\nu j})!} \,, \tag{5}$$

where the factorial of the number operator denotes the operator product $\hat{n}! = \hat{n}(\hat{n}-1)\dots$ The combinatoric factor ensures that the rate is proportional to the current number of groups of

each type of agent needed for the reaction (i.e. the number of groups of $m_1$ of agent $A_1$ and so on).

The finite time evolution of probability distributions is governed by a time evolution operator, $\hat{\mathcal{U}}_t = \exp(t\hat{\mathcal{W}})$. Reaction models with stable dynamics drive initial states toward some stationary state at long times. Stationary states $\mathcal{P}_{\mathrm{st}}$ are the eigenvalues of $\hat{\mathcal{W}}$ with zero eigenvalue, $\hat{\mathcal{W}} |\mathcal{P}_{\mathrm{st}}\rangle = 0$. With this machinery, statistics of population models are computed as the expectation values of number operators inserted between time evolution operators,

$$\langle n_{j_N}(t_N) \dots n_{j_2}(t_2) n_{j_1}(t_1) \rangle_{\mathcal{P}} = \langle 1| \hat{n}_{j_N} \hat{\mathcal{U}}_{t_N - t_{N-1}} \dots \hat{n}_{j_2} \hat{\mathcal{U}}_{t_2 - t_1} \hat{n}_{j_1} \hat{\mathcal{U}}_{t_1} |\mathcal{P}\rangle \,. \tag{6}$$

When expectations are taken in the stationary state, the subscript $\mathcal{P}$ will be omitted.

An absorbing state $\mathcal{P}_{\mathrm{abs}}$ is a stationary state which, once entered, cannot be left. Such a state must be inert under each reaction process in the model independently, $\hat{\mathcal{W}}_\nu |\mathcal{P}_{\mathrm{abs}}\rangle = 0$ for all $\nu$. Expectations in absorbing states are independent of all time arguments,

$$\langle n_{j_N}(t_N) \dots n_{j_2}(t_2) n_{j_1}(t_1) \rangle = \langle n_{j_N} \dots n_{j_2} n_{j_1} \rangle = \bar{n}_{j_N} \dots \bar{n}_{j_2} \bar{n}_{j_1} \,, \tag{7}$$

where the final equality holds for absorbing "pure" states, with $|\mathcal{P}_{\mathrm{abs}}\rangle = |\bar{\mathbf{n}}\rangle$. Such states are totally *fluctuationless*: there is no uncertainty and all expectations reduce to products of $\bar{n}_j$. A reaction model can be interpreted as a population dynamics when the state of zero population of all species $|0\rangle$ is an absorbing state: animals may be born and die, but they can never emerge from a state with no initial population. Such a state represents total extinction, so the population has no fluctuations and all statistical moments must be zero. On the level of the reaction rules Eq. (1), all processes must have $m_j > 0$ for at least one species $j$.

We consider now a set of prototypical reaction rules to exemplify the ideas discussed above. For simplicity, consider a model with only a single specie $A$. Some simple reaction rules and their corresponding evolution operators are:

$$A \xrightarrow{\gamma_{\mathrm{d}}} \emptyset \qquad \longleftrightarrow \qquad \gamma_{\mathrm{d}} \hat{\mathcal{W}}_{\mathrm{d}} = \gamma_{\mathrm{d}} \left( e^{-\hat{p}} - 1 \right) \hat{n} \,, \tag{8a}$$

$$A \xrightarrow{\gamma_{\mathrm{r}}} 2A \qquad \longleftrightarrow \qquad \gamma_{\mathrm{r}} \hat{\mathcal{W}}_{\mathrm{r}} = \gamma_{\mathrm{r}} \left( e^{\hat{p}} - 1 \right) \hat{n} \,, \tag{8b}$$

$$\emptyset \xrightarrow{\gamma_{\mathrm{s}}} A \qquad \longleftrightarrow \qquad \hat{\gamma}_{\mathrm{s}} \hat{\mathcal{W}}_{\mathrm{s}} = \gamma_{\mathrm{s}} \left( e^{\hat{p}} - 1 \right) \,. \tag{8c}$$

Both of the first two reaction rules disallow transitions out of the extinct state and are thus valid processes in a population dynamics, respectively representing death and (asexual) reproduction of animals. This is formally ensured by the proportionality to $\hat{n}$, as $\hat{n} |0\rangle = 0$. The third rule represents a "spontaneous generation" of animals out of the extinct state, and so is *not* allowed in population dynamics models.

The two terms in the brackets $(e^{\pm\hat{p}} - 1)$ in Eqs. (8) represent "in" and "out" processes. For example, for the death process of Eq. (8a) the "in" process is initiated in a state with $n + 1$ animals, which after death of one of them leads to the state $|n\rangle$; the "out" process leads from the state $|n\rangle$ to a state with $n - 1$ animals. This is achieved by acting $\left( e^{-\hat{p}} - 1 \right) \hat{n} |\mathcal{P}\rangle = \sum_n \mathcal{P}_n \left( e^{-\hat{p}} - 1 \right) \hat{n} |n\rangle = \sum_n \mathcal{P}_n n \left( |n-1\rangle - |n\rangle \right) = \sum_n \left( (n+1)\mathcal{P}_{n+1} - n\mathcal{P}_n \right) |n\rangle$. As a result, equation (4) spells out the following evolution law: $\partial_t \mathcal{P}_n = \gamma_{\mathrm{d}} \left( (n+1)\mathcal{P}_{n+1} - n\mathcal{P}_n \right)$, which indeed represents $A \xrightarrow{\gamma_{\mathrm{d}}} \emptyset$ reaction.

Reaction models are naturally extended to finite dimensional settings by assigning populations of each type of agents to sites on a lattice, $A_{j\mathbf{r}}$ where $\mathbf{r}$ is a lattice vector. With this, reaction rules can also represent the movement of agents to different points in space. The simplest example to this is the combination of the two rules $A_{\mathbf{r}} \to A_{\mathbf{r} \pm \mathbf{a}_i}$ for all lattice unit vectors $\mathbf{a}_i$. This implements a random walk on the lattice which causes diffusion of the population.

## 2.2 Lindbladians and superoperators

We review the basics of the superoperator formalism, as it is most naturally related to the operator treatment of classical reaction models presented above. The Lindblad equation describes the Markovian evolution of the reduced density matrix *operator*, $\hat{\rho}$, of an open quantum system. The Lindbladian superoperator acts on the space of quantum operators. Introducing doubled bra-ket notation, we take $\hat{O} = |O\rangle\rangle$ to denote a vector in the space of operators acting on the quantum Hilbert space of wave functions. The Lindblad equation may then be expressed as:

$$\partial_t |\rho\rangle\rangle = \hat{\mathcal{L}} |\rho\rangle\rangle, \tag{9a}$$

$$\hat{\mathcal{L}} = -i\hat{\mathcal{H}} + \sum_\nu \gamma_\nu \hat{\mathcal{D}}[\hat{L}_\nu]. \tag{9b}$$

The superoperators $\hat{\mathcal{H}}$ and $\hat{\mathcal{D}}$ respectively represent Hamiltonian and dissipative contributions to the dynamics. Letting $\hat{O}_+ |\rho\rangle\rangle = \hat{O}\hat{\rho}$ and $\hat{O}_- |\rho\rangle\rangle = \hat{\rho}\hat{O}$ be superoperators that denote the left and right action of $\hat{O}$, these are:

$$\hat{\mathcal{H}} = \hat{H}_+ - \hat{H}_-, \tag{10a}$$

$$\hat{\mathcal{D}}[\hat{L}] = \hat{L}_+ \hat{L}_-^\dagger - \frac{1}{2}\left(\hat{L}_+^\dagger \hat{L}_+ + \hat{L}_- \hat{L}_-^\dagger\right). \tag{10b}$$

Here, $\hat{H}$ the system Hamiltonian and $\hat{L}_\nu$ the jump operators, which govern incoherent processes caused by interaction with the environment; $\gamma_\nu > 0$ is a set of real positive coupling constants between the quantum system and its environment(s). Letting $\hat{a}_j$ denote a set of $N$ bosonic modes, with $[\hat{a}_i, \hat{a}_j^\dagger] = \delta_{ij}$, we consider $\hat{H}$ and $\hat{L}_\nu$ to be polynomials in $\hat{a}_j$ and $\hat{a}_j^\dagger$.

The time evolution of the density matrices is controlled by the time evolution superoperator $\hat{\mathcal{U}}_t = \exp(t\hat{\mathcal{L}})$. At long times the density matrix is driven toward a stationary state, $\hat{\rho}_{st}$, which is the eigenvalues of $\hat{\mathcal{L}}$ that have zero eigenvalue, $\hat{\mathcal{L}} |\rho_{st}\rangle\rangle = 0$. It is natural to equip the operator space with the Hilbert-Schmidt norm, $\langle\langle O|O'\rangle\rangle = \text{tr}(\hat{O}^\dagger \hat{O}')$. With this, the doubled bra $\langle\langle 1|$ can be used to take a trace, allowing computation of expectations via the insertion of operators on either side of the density matrix. As an example, generic $N$-point functions are given by:

$$\langle\langle 1|\hat{a}_{j_N \sigma_N} \hat{\mathcal{U}}_{t_N - t_{N-1}} \cdots \hat{a}_{j_2 \sigma_2}^\dagger \hat{\mathcal{U}}_{t_2 - t_1} \hat{a}_{j_1 \sigma_1} \hat{\mathcal{U}}_{t_1} |\rho\rangle\rangle, \tag{11}$$

where $\sigma \in \{+, -\}$, and similarly for any string of either $\hat{a}$ or $\hat{a}^\dagger$.

## 2.3 Weak symmetry and dark states

Open quantum systems have two different notions of symmetry [15,66,67]. A strong symmetry is a typical symmetry of a quantum system. In contrast, a *weak* symmetry is a symmetry of the system and environment jointly, but which holds only on average for the effective Lindbladian dynamics of the system alone. Weak symmetries do not correspond to conservation laws, but they do have consequences on dynamics.

In the present context, we consider $N$ copies of $U(1)$. To define this, we first introduce the bosonic number and phase operators,

$$\hat{a}_j = e^{i\hat{\theta}_j}\sqrt{\hat{n}_j}, \qquad \hat{a}_j^\dagger = \sqrt{\hat{n}_j}\, e^{-i\hat{\theta}_j}. \tag{12}$$

The operators $\hat{n}_j$ and $\hat{\theta}_j$ are Hermitian and satisfy the canonical commutation relations $[\hat{n}_j, \hat{\theta}_i] = i\delta_{ij}$. These are the same as the operator representation used for the classical population model as defined in Eqs. (2) and (3), with $\hat{p}_j = i\hat{\theta}_j$. With this, we define the $U(1)$ symmetry operator $\hat{U}_j(\vartheta) = \exp(-i\hat{n}_j \vartheta)$ and corresponding superoperator $\hat{\mathcal{U}}_j(\vartheta) = \hat{U}_{j+}(\vartheta)\hat{U}_{j-}^\dagger(\vartheta)$.

A Lindbladian is weakly symmetric if

$$[\hat{\mathcal{L}}, \hat{\mathcal{U}}_j(\vartheta)] = 0. \tag{13}$$

As an example consider a single bosonic mode with the jump operators $\hat{L}_\nu = \hat{a}, \hat{a}^\dagger$. They transform respectively as $\hat{\mathcal{U}}(\vartheta)|a\rangle\rangle = e^{-i\hat{n}\vartheta}\hat{a}\,e^{i\hat{n}\vartheta} = e^{i\vartheta}\hat{a}$ and $\hat{\mathcal{U}}(\vartheta)|a^\dagger\rangle\rangle = e^{-i\vartheta}\hat{a}^\dagger$. Examining the form of the dissipative superoperator in Eq. (10) it is clear that an overall phase of $\hat{L}$ does not affect the form of $\hat{\mathcal{D}}$, and thus one has $\hat{\mathcal{D}}[e^{-i\vartheta}\hat{a}^\dagger] = \hat{\mathcal{D}}[\hat{a}^\dagger]$ and $\hat{\mathcal{D}}[e^{i\vartheta}\hat{a}] = \hat{\mathcal{D}}[\hat{a}]$. This implies the invariance of the overall Lindbladian under the weak symmetry transformation. This fails for jump operators that are not invariant up to an overall phase under the symmetry; for example $\hat{L} = \hat{a} + \hat{a}^\dagger$ which leads to a Lindbladian that is not invariant under symmetry $\hat{\mathcal{D}}[e^{i\vartheta}\hat{a} + e^{-i\vartheta}\hat{a}^\dagger] \neq \hat{\mathcal{D}}[\hat{a} + \hat{a}^\dagger]$.

This idea can be generalized to establish conditions on the Hamiltonian and jump operators that lead to weak symmetry. A charged operator $\hat{O}$ with charge $q_O$ is an eigenvector of the $U(1)$ symmetry superoperator $\hat{\mathcal{U}}(\vartheta)|O\rangle\rangle = e^{iq_O\vartheta}|O\rangle\rangle = e^{iq_O\vartheta}\hat{O}$. A neutral operator is a charged operator with charge 0 for all $U(1)$ factors. The Hamiltonian part of the Lindbladian respects weak symmetry when the Hamiltonian is symmetric under each factor of $U(1)$ in the conventional sense, i.e. when it is neutral $\hat{\mathcal{U}}_j(\vartheta)|H\rangle\rangle = |H\rangle\rangle = \hat{H}$. The dissipative part is weakly symmetric when each jump operator is charged under every copy of $U(1)$.[1] In the above example with the single bosonic mode, $\hat{a}$ and $\hat{a}^\dagger$ are seen to have charges 1 and -1 respectively, while the operator $\hat{a} + \hat{a}^\dagger$ does not have a definite charge (i.e. is not charged) and thus does not lead to a symmetric Lindbladian.

Weak continuous symmetry does *not* conserve charge. However, weakly symmetric dynamics may still be decomposed into different charge sectors. A state with a definite charge may gain or lose charge, but it will never evolve to a quantum superposition of states with different charges. To express this idea formally, it is convenient to introduce the "classical" and "quantum" number and phase superoperators,

$$\hat{n}^c = (\hat{n}_+ + \hat{n}_-)/2, \qquad \hat{n}^q = \hat{n}_+ - \hat{n}_-, \qquad \hat{\theta}^c = (\hat{\theta}_+ + \hat{\theta}_-)/2, \qquad \hat{\theta}^q = \hat{\theta}_+ - \hat{\theta}_-, \tag{14a}$$

$$[\hat{n}_j^c, \hat{\theta}_i^q] = i\delta_{ij} = [\hat{n}_j^q, \hat{\theta}_i^c], \qquad [\hat{n}_j^\alpha, \hat{\theta}_i^\alpha] = 0, \qquad [\hat{n}_j^\alpha, \hat{n}_i^\beta] = 0 = [\hat{\theta}_j^\alpha, \hat{\theta}_i^\beta], \tag{14b}$$

with $\alpha, \beta \in \{c, q\}$. The operator Hilbert space is spanned by the Fock state operators $|\mathbf{n}\rangle\langle\mathbf{n}'| = |\mathbf{n}^c, \mathbf{n}^q\rangle\rangle$, where $\mathbf{n}^c = (\mathbf{n} + \mathbf{n}')/2$ and $\mathbf{n}^q = \mathbf{n} - \mathbf{n}'$. The operators $|\mathbf{n}\rangle\langle\mathbf{n}'|$ are eigenvectors of the classical and quantum number superoperators; the phase superoperators define raising and lowering operators,

$$\hat{n}_j^c|\mathbf{n}^c, \mathbf{n}^q\rangle\rangle = n_j^c|\mathbf{n}^c, \mathbf{n}^q\rangle\rangle, \qquad \hat{n}_j^q|\mathbf{n}^c, \mathbf{n}^q\rangle\rangle = n_j^q|\mathbf{n}^c, \mathbf{n}^q\rangle\rangle, \tag{15a}$$

$$\exp\left(i\hat{\theta}_j^q\right)|\mathbf{n}^c, \mathbf{n}^q\rangle\rangle = |\ldots, n_j^c + 1, \ldots, \mathbf{n}^q\rangle\rangle, \qquad \exp\left(i\hat{\theta}_j^c\right)|\mathbf{n}^c, \mathbf{n}^q\rangle\rangle = |\mathbf{n}^c, \ldots, n_j^q + 1, \ldots\rangle\rangle. \tag{15b}$$

Note that $n_j^c$ is a positive half-integer and $n_j^q \in \{-2n_j^c, -2n_j^c + 2, \ldots 2n_j^c\}$.

The weak $U(1)$ symmetry superoperator may be expressed $\hat{\mathcal{U}}_j(\vartheta) = \exp(i\hat{n}_j^q\vartheta)$. Thus, a Lindbladian is weakly symmetric when $[\hat{\mathcal{L}}, \hat{n}_j^q] = 0$. As a consequence $\mathbf{n}^q$ is "conserved" and is thus a good quantum number for the Lindbladian eigenspace. The quantum number $\mathbf{n}^q$ specifies the off-diagonal character of operators in the number basis. The operators with $\mathbf{n}^q = 0$ are diagonal and thus represent statistical mixtures of states with definite numbers of particles, i.e. $|\mathbf{n}, 0\rangle\rangle = |\mathbf{n}\rangle\langle\mathbf{n}|$. In contrast, eigenvectors with $\mathbf{n}^q \neq 0$ involve quantum superpositions of

---

[1]This is straight-forwardly proved by generalizing the argument used in the example: the dissipative super-operator is independent of the phase of $\hat{L}$, $\hat{\mathcal{D}}[e^{i\alpha}\hat{L}] = \hat{\mathcal{D}}[\hat{L}]$. It is straightforward to check by action on $|\rho\rangle\rangle$ that $\hat{\mathcal{U}}(\vartheta)\hat{\mathcal{D}}[\hat{L}] = \hat{\mathcal{D}}[\hat{\mathcal{U}}(\vartheta)\hat{L}]\hat{\mathcal{U}}(\vartheta)$. Thus, for $\hat{L}$ charged, $\hat{\mathcal{U}}(\vartheta)\hat{L} = e^{iq_L\vartheta}\hat{L}$, it follows that $[\hat{\mathcal{U}}(\vartheta)\hat{\mathcal{D}}[\hat{L}], \hat{\mathcal{D}}[\hat{L}]\hat{\mathcal{U}}(\vartheta)] = 0$.

different numbers of particles. With weak symmetry, the $\mathbf{n}^{\mathrm{q}}$ sectors of the eigenspace do not mix and thus quantum superpositions are never generated from initially classical states.

This makes the comparison of the weak symmetry Lindbladian dynamics to the classical reaction models immediate: the Lindbladian dynamics of the $\mathbf{n}^{\mathrm{q}} = 0$ sector is exactly equivalent to some classical reaction model. The operator double ket $|\mathbf{n}, 0\rangle\rangle = |\mathbf{n}\rangle \langle \mathbf{n}|$ corresponds exactly to the classical state $|\mathbf{n}\rangle$ and the diagonal entries of the density matrix $\langle \mathbf{n}|\hat{\rho}|\mathbf{n}\rangle$ correspond to the classical probability distribution $\mathcal{P}_{\mathbf{n}}$. This mapping is *exact* on the level of operators: for $\mathbf{n}^{\mathrm{q}} = 0$ the Hamiltonian part of the Lindbladian always vanishes[2] and the dissipative part maps to a classical reaction model, where $\hat{n}^{\mathrm{c}}_j$ and $-\mathrm{i}\hat{\theta}^{\mathrm{q}}_j$ in the quantum model correspond to $\hat{n}_j$ and $\hat{p}_j$ in the classical model and $\hat{\theta}^{\mathrm{c}}$ never appears in symmetric Lindbladians. Conversely, *every classical reaction rule dynamics model may be represented as a pure dissipative Lindbladian evolution, obeying a set of weak $U(1)$ symmetries, which fix $\mathbf{n}^{\mathrm{q}} = 0$ as a conserved quantity.*

To illustrate how this works, consider the example with the jump operators $\hat{L}_\nu = \hat{a}, \hat{a}^\dagger$. Then their corresponding dissipative superoperators may be expressed in terms of number and phase superoperators. Using the Eq. (12), one finds for the bosonic superoperators:

$$\hat{a}_+ = \exp\left(\mathrm{i}\hat{\theta}_+\right)\sqrt{\hat{n}_+}, \qquad \hat{a}_- = \sqrt{\hat{n}_-}\exp\left(\mathrm{i}\hat{\theta}_-\right). \tag{16}$$

Then by making use of the classical and quantum number operators and their commutation rules, one can bring the dissipative superoperator to "normal-order" in which all of the number superoperators are to the right of the phase superoperators. The result is:

$$\hat{\mathcal{D}}[\hat{a}] = \exp\left(\mathrm{i}\hat{\theta}^{\mathrm{q}}\right)\sqrt{\hat{n}^{\mathrm{c}2} - \frac{1}{4}\hat{n}^{\mathrm{q}2}} - \hat{n}^{\mathrm{c}}, \tag{17a}$$

$$\hat{\mathcal{D}}[\hat{a}^\dagger] = \exp\left(-\mathrm{i}\hat{\theta}^{\mathrm{q}}\right)\sqrt{(\hat{n}^{\mathrm{c}} + 1)^2 - \frac{1}{4}\hat{n}^{\mathrm{q}2}} - (\hat{n}^{\mathrm{c}} + 1). \tag{17b}$$

Upon putting $n^{\mathrm{q}} = 0$ and exchanging $\hat{n}^{\mathrm{c}}$ and $-\mathrm{i}\hat{\theta}^{\mathrm{q}}$ for $\hat{n}$ and $\hat{p}$, these match *exactly* the expressions in Eq. (8):

$$\hat{\mathcal{D}}[\hat{a}]\big|_{n^{\mathrm{q}}=0} = \left(\exp\left(\mathrm{i}\hat{\theta}^{\mathrm{q}}\right) - 1\right)\hat{n}^{\mathrm{c}} \longleftrightarrow (\mathrm{e}^{-\hat{p}} - 1)\hat{n} = \hat{\mathcal{W}}_{\mathrm{d}}, \tag{18a}$$

$$\hat{\mathcal{D}}[\hat{a}^\dagger]\big|_{n^{\mathrm{q}}=0} = \left(\exp\left(-\mathrm{i}\hat{\theta}^{\mathrm{q}}\right) - 1\right)(\hat{n}^{\mathrm{c}} + 1) \longleftrightarrow (\mathrm{e}^{\hat{p}} - 1)(\hat{n} + 1) = \hat{\mathcal{W}}_{\mathrm{r}} + \hat{\mathcal{W}}_{\mathrm{s}}. \tag{18b}$$

Thus, the diagonal dynamics of $\hat{L} = \hat{a}$ is equivalent to the classical reaction rule $A \to \emptyset$ and $\hat{L} = \hat{a}^\dagger$ is equivalent to the combination $A \to 2A$ and $\emptyset \to A$ occurring at the same rate. The correspondence between other (weakly symmetric) jump operators and classical reaction rules can be determined using the number representation in a similar way. In general single jump operators correspond to several reaction rules simultaneously. Some examples are listed in Table 1.

Like classical reaction models, Lindbladian dynamics can feature stationary states which cannot be exited once entered. We will use the term dark state for such quantum absorbing states to clearly distinguish with the classical setting. Formally, a dark state, $\hat{\rho}_{\mathrm{D}}$, is a stationary state which is annihilated by both the Hamiltonian, $\hat{\mathcal{H}}$, part and *each* dissipative term, $\hat{\mathcal{D}}[\hat{L}_\nu]$, of the Lindbladian independently. For a *pure* dark state, $\hat{\rho}_{\mathrm{D}} = |\mathrm{D}\rangle \langle \mathrm{D}|$, state $|\mathrm{D}\rangle$ must be an eigenvector of $\hat{H}$ and must be annihilated by every jump operator, $\hat{L}_\nu |\mathrm{D}\rangle = 0$. In such cases,

---

[2]For polynomials of $\hat{a}_j$ and $\hat{a}^\dagger_j$, symmetry requires $\hat{H}$ to depend only on the number $\hat{n}_j$ and not the phase $\hat{\theta}_j$ of each bosonic mode. All terms in the Hamiltonian superoperator $\hat{\mathcal{H}}$ are thus always proportional to at least one factor of $\hat{n}^{\mathrm{q}}_j$, as follows from the requirement that $\hat{\mathcal{H}}$ preserve the density matrix trace.

Table 1: The right column shows the classical reaction rules describing a classical dynamics of the charge diagonal subspace of the corresponding jump operators in the left column. The rate $\gamma$ is the rate of the quantum process, so that the dissipative part of the Lindbladian is $\gamma \hat{\mathcal{D}}[\hat{L}]$. The crossed out entry corresponding to $\hat{L} = \hat{a}^\dagger \hat{a}$ indicates that there are no reaction rules for this jump operator. This exemplifies the more general fact that neutral operators have no effect on the *diagonal* subspace and thus do not correspond to any classical reaction rules (they do nevertheless have an effect on the overall dynamics, causing decoherence of the $\mathbf{n}^q \neq 0$ subspaces). One should also make special note of the second to last entry, in which the jump operator translates to classical reaction rules even though it is sum of terms with different numbers of bosonic creation operators because each term has the same charge.

| Jump Operator $\hat{L}$ | Reaction Rules |
|:---:|:---:|
| $\hat{a}$ | $A \xrightarrow{\gamma} \emptyset$ |
| $\hat{a}^\dagger$ | $\emptyset \xrightarrow{\gamma} A, A \xrightarrow{\gamma} 2A$ |
| $\hat{a}^2$ | $2A \xrightarrow{2\gamma} \emptyset$ |
| $\hat{a}^\dagger \hat{a}$ | $\times$ |
| $\hat{a}^\dagger \hat{a}^2$ | $3A \xrightarrow{6\gamma} 2A, 2A \xrightarrow{2\gamma} A$ |
| $\hat{a}^{\dagger 2} \hat{a}$ | $3A \xrightarrow{6\gamma} 4A, 2A \xrightarrow{8\gamma} 3A, A \xrightarrow{2\gamma} 2A$ |
| $(\hat{a}^\dagger \hat{a} + 1)\hat{a}$ | $3A \xrightarrow{6\gamma} 2A, 2A \xrightarrow{6\gamma} A, A \xrightarrow{\gamma} \emptyset$ |
| $\hat{b}^\dagger \hat{a}$ | $A \xrightarrow{\gamma} B, A + B \xrightarrow{\gamma} 2B$ |

any set of observables $\{\hat{O}_a\}$ with $|D\rangle$ as an eigenvector, $\hat{O}_a |D\rangle = O_a^{(D)} |D\rangle$, have no fluctuations for any time ordering,

$$\langle \hat{O}_{a_N}^{\sigma_N}(t_N) \cdots \hat{O}_{a_2}^{\sigma_2}(t_2) \hat{O}_{a_1}^{\sigma_1}(t_1) \rangle = O_{a_N}^{(D)} \cdots O_{a_2}^{(D)} O_{a_1}^{(D)}. \tag{19}$$

This is analogous to the classical fluctuationlessness as in Eq. (7). Unlike the classical case, though, this does not apply for all expectation values: operators that do not have $|D\rangle$ as an eigenvector, still have correlation functions with a nontrivial time dependence.

## 2.4 Quantum population dynamics

A weakly symmetric Lindbladian dynamics which has the Fock vacuum, $|0\rangle$, as a dark state has diagonal dynamics equivalent to a classical population dynamics. We thus define a *quantum population dynamics* as a Lindbladian dynamics that has the Fock vacuum as a dark state but is *not* required to be weakly symmetric. This retains intuition of the zero population state being an absorbing, fluctuationless state, while allowing for distinctly quantum processes in which states with definite numbers of animals may evolve to quantum superpositions of different numbers. In such models, all jump operators have the form $\hat{L}_\nu = \hat{O}_\nu \hat{a}_j$ where $\hat{O}_\nu$ can be any polynomial of $\hat{a}_j$ and $\hat{a}_j^\dagger$. The Lindbladian eigenspace mixes charge sectors and $\mathbf{n}^q$ is no longer a good quantum number. As a consequence, generically all other eigenvectors, possibly including other stationary states, involve superpositions of different populations. Note that for such models, a stronger statement than Eq. (19) also holds: any time-ordered dark state expectations of any charged operators vanish identically.

As a simple illustration of this concept we introduce what we dub a "Schrödinger cat" process, in which an animal is driven into a coherent superposition of being alive and dead: $|1\rangle \rightarrow |1\rangle + |0\rangle$. One choice of jump operator that encodes such process is $\hat{L}_c = \hat{a}^\dagger \hat{a} + \hat{a}$, which acts as:

$$\hat{L}_c |n\rangle = n |n\rangle + \sqrt{n} |n-1\rangle . \tag{20}$$

Thus, such (intrinsically incoherent) processes take classical states of a given population to coherent quantum superpositions of different population numbers.

In higher spatial dimensions, in order for a Lindbladian to have a classical diagonal dynamics the weak symmetry must be promoted to a *local* symmetry, $[\hat{\mathcal{U}}_j(\vartheta_\mathbf{r}), \hat{\mathcal{L}}] = 0$, with $\mathbf{r}$ a lattice vector. This condition is compatible with incoherent hopping terms, $\hat{L}_{\mathbf{r}, \pm \mathbf{a}_i} = \hat{a}^\dagger_{\mathbf{r} \pm \mathbf{a}_i} \hat{a}_\mathbf{r}$, which when taken together with each lattice unit vector $\mathbf{a}_i$ define a density dependent classical random walk as a combination of the processes $A_\mathbf{r} \rightarrow A_{\mathbf{r}+\mathbf{a}_i}$ and $A_\mathbf{r} + A_{\mathbf{r}+\mathbf{a}_i} \rightarrow 2A_{\mathbf{r}+\mathbf{a}_i}$ (see Table 1). However, local symmetry is not compatible with more standard kinds of quantum hopping terms, even those respecting global $U(1)$. Thus, quantum hopping either in the Hamiltonian or in a jump operator is an alternative way to introduce quantum effects in the population dynamics. As an example, consider dissipation on the bond of the lattice $\hat{L}_{\mathbf{r}, \pm \mathbf{a}_i,} = \hat{a}_{\mathbf{r} \pm \mathbf{a}_i} + \hat{a}_\mathbf{r}$. In the simplest case of a 1D chain, this acts as:

$$\hat{L}_{r,+} |\ldots, n_r, n_{r+1}, \ldots\rangle = |\ldots, n_r, n_{r+1} - 1, \ldots\rangle + |\ldots, n_r - 1, n_{r+1}, \ldots\rangle . \tag{21}$$

Such processes do not mix states with different amounts of total charge, but instead generate superpositions of populations at different points in space.

Classical population models may have more than one absorbing state. One example is provided by a set of reactions in which all processes require at least $k > 1$ agents to initiate a reaction [68]. In such multi-agent processes, any probability distribution over only the states involving less than $k$ animals is an absorbing state. A natural quantum generalization of the multi-agent process can be defined by only allowing jump operators of the form $\hat{L}_\nu = \hat{O}_\nu \hat{a}^k$, which has a space of dark states spanned by the Fock states $|n\rangle$, $n < k$. A potentially more interesting appropriation of the spirit of such models is to instead build a model with a dark space that is quantum in nature. As an example, consider the jump operator

$$\hat{L}_{\phi_0} = \hat{a}^2 - \phi_0 \hat{a} , \tag{22}$$

where $\phi_0$ is a complex number. It can be thought of as a superposition of the two classical processes $2A \rightarrow \emptyset$ and $A \rightarrow \emptyset$. In addition to the Fock vacuum, $|0\rangle$, $\hat{L}_{\phi_0}$ also annihilates the coherent state $|\phi_0\rangle$ and therefore possesses a dark *space* spanned by the two states $|\phi_0\rangle$ and $|0\rangle$. Thus, despite being composed of two classical processes which both act to reduce the overall population, such a quantum population process stabilizes a state which, for large $|\phi_0|$, may have a large average population. This property is retained so long as all jump operators are of the form $\hat{L}_{\phi_0 \nu} = \hat{O}_\nu (\hat{a} - \phi_0) \hat{a}$. It can also be retained in finite dimensional models, for example by including quantum diffusion terms like Eq. (21). Because such terms also kill the spatially homogeneous coherent state, such models may thus feature macroscopically large dark populations even in the thermodynamic limit.

For the remainder of this manuscript we will focus on generic quantum population dynamics, without any additional symmetries or constraints. The quantum population dynamic rules with extra constraints or symmetries, which may include presence of dark spaces, are left for future inquiries.

# 3 Field theory of absorbing states

In zero dimensions and in the absence of additional constraints or symmetries a stable classical population dynamics always has only one true stationary state: the dead absorbing state of a total extinction. Even in models with competition between proliferation and death which may lead to long-lived meta-stable states with finite populations, there is always a finite extinction probability due to rare events. In spatially extended models, areas of parameter space which support meta-stable states are extended into dynamical phases with stable (but fluctuating) populations. In such active phases the extinct absorbing state is dynamically inaccessible: initial conditions with small regions of finite population grows, eventually leading to a stable finite population across the entire space. Active phases and dead phases, in which all initial populations eventually die, are separated by a continuous phase transition. The simplest example of such classical absorbing state transition with a single species and no additional discrete symmetries or constraints belongs to the directed percolation universality class [18,46–49,52].

This naturally poses the question as to whether quantum generalizations of population models, as introduced in the previous section, may feature new types of continuous transitions between an inactive dark phase and a phase with a dynamically inaccessible dark state. In order to address this question, we develop a field theory treatment of (bosonic) quantum population models.

## 3.1 Classical directed percolation

The most basic example of a classical absorbing state transition with a single species and no additional discrete symmetries or constraints belongs to the directed percolation universality class [18, 46–49]. We will not present derivation, which can be done for example using Doi-Peliti formalism [44, 45].

The effective stochastic field theory of a classical population dynamics in $d$ spatial dimensions is encoded in the partition function:

$$Z = \int Dn\, Dp\, e^{-S[n,p]}, \tag{23a}$$

$$S[n,p] = \int dx \left( p(\partial_t - D\partial_{\mathbf{r}}^2)n - W(n,p) \right), \tag{23b}$$

where $x = (t, \mathbf{r})$, $n(x)$ is the local population number, and $p(x)$ is a canonically conjugated auxiliary field. The effective local "Hamiltonian", $W(n, p)$, is obtained from the corresponding *normally ordered* reaction operators, $\hat{\mathcal{W}}(\hat{n}, \hat{p})$, of, e.g., Eqs. (8a,b) by substituting canonically conjugated operators $\hat{n}$ and $\hat{p}$ with functions of time (and space), $n(x)$ and $p(x)$, [52]. Expanding such obtained $W(n, p)$ to the lowest non-trivial orders in both $n$ and $p$, one arrives at:

$$W(n,p) = p(\delta - g_1 n + g_2 p)n. \tag{24}$$

This defines the action for the so-called Reggeon field theory [69–71]. Many different reaction rules can give rise to the same long wavelength theory. One such combination of reaction processes is death, asexual reproduction, and competition: $A \to \emptyset$, $A \to 2A$, and $2A \to \emptyset$. Diffusion can be obtained by the random walk $A_{\mathbf{r}} \to A_{\mathbf{r} \pm \mathbf{a}_i}$ on, for example, a square lattice. Any processes involving larger groups of animals can also be added, but they will not effect the form of the leading-order parts of the action. Such additional higher-order terms may be included, but they are irrelevant in the RG sense and so will have minimal effects on the universal features of the model.

The noiseless equation of motion for the local number is the well-known FKPP equation [72, 73]:

$$(\partial_t - D\partial_{\mathbf{r}}^2)n = (\delta - g_1 n)n. \tag{25}$$

The mass term $\delta$ microscopically originates from the difference between the birth and death rates and thus may be either positive or negative. This predicts two distinct phases, which at the mean field level depends on the sign of $\delta$. For $\delta < 0$, i.e. when the death rate exceeds the birth rate, there are no stable solutions other than $n = 0$. This indicates an absorbing phase, in which all populations eventually die and the only stationary state is total extinction. Alternatively for $\delta > 0$ there is an active phase: any nonzero population grows until reaching the environmental capacity, eventually saturating to a stable average value $\langle n \rangle = \delta/g_1$.

The Reggeon field theory constitutes a universal theory for classical absorbing state transitions in the absence of additional symmetries or dynamical constraints. Like all stochastic field theories the action obeys $S[n, p = 0] = 0$, which is required for the conservation of probability. As a consequence of the absorbing state, it also possesses a parallel property for the classical field,

$$S[n = 0, p] = 0. \tag{26}$$

This property guarantees the validity of Eq. (7), which in the present context implies:

$$\langle n(x_N) \cdots n(x_2)n(x_1) \rangle = 0. \tag{27}$$

That is, *all* correlation functions of only the classical field $n$ are identically zero in the absorbing state. This is an exact property of the action and is present for all microscopic population models regardless of what particular reaction rules are present. In equilibrium, symmetries determine the content of a theory and the universality of any phase transitions by constraining what terms are permitted in the action and consequently what types of RG flows are allowed. In reaction models with absorbing states, it is instead dynamical constraints which put conditions on the action (in the present context, $S[n = 0, p] = 0$) that specify what terms are allowed. The requirement that such conditions be preserved under RG lead to universality distinct from those present in equilibrium models, even in the absence of symmetry.

The Reggeon field theory happens to possesses the Reggeon inversion symmetry, which is a $\mathbb{Z}_2$ symmetry that combines time reversal and the exchange of the auxiliary and classical fields:

$$n(t, \mathbf{r}) \rightarrow -\frac{g_2}{g_1}p(-t, \mathbf{r}), \qquad p(t, \mathbf{r}) \rightarrow -\frac{g_1}{g_2}n(-t, \mathbf{r}). \tag{28}$$

This symmetry is not always present microscopically, but does emerge at long wavelengths for many different population models. This provides another characterization of the difference between the absorbing and active phases: the symmetry is spontaneously broken in the active phase, with the finite population, $\langle n \rangle$, acting as $\mathbb{Z}_2$ order parameter. We note that symmetry breaking is not required for all absorbing state transitions and indeed no such symmetry is present in other types of absorbing state transitions [18, 49, 50], for example those possessing additional symmetries or dynamical constraints.

The scaling of various quantities near the critical point can be studied using RG. The resulting critical exponents define the directed percolation universality class. The upper critical dimension of the model is $d = 4$. Using a standard $\epsilon$-expansion approach with $\epsilon = 4 - d$, one finds Wilson-Fisher fixed point with the critical exponents to leading order in $\epsilon$ given by [18, 46–49, 52]:

$$z = 2 - \frac{\epsilon}{12}, \qquad \nu = \frac{1}{2} + \frac{\epsilon}{16}, \qquad \alpha = 1 - \frac{\epsilon}{4}, \qquad \beta = 1 - \frac{\epsilon}{6}, \tag{29}$$

where $z$ is the dynamical critical exponent, $\nu$ is the correlation length exponent, $\alpha$ sets the extinction dynamics right at the criticality $\langle n(t) \rangle \sim t^{-\alpha}$, and $\beta$ determines the critical scaling of the average population size near the critical point $\langle n \rangle \sim \delta^\beta$ for $\delta > 0$.

## 3.2 Keldysh formalism for theories with dark states

To study the critical behaviour of quantum population models, a similar field theory treatment is needed. The number and phase representation introduced in the previous section is not easily amenable to a field theory representation, so we instead develop the standard coherent state Keldysh path integral for this purpose [15, 17, 52]. For simplicity we consider a single bosonic mode in $d$ dimensions. We focus on quantum population models, in which the Fock vacuum $|0\rangle$ is a dark state of the Lindbladian dynamics. This requirement constrains the allowed terms in the action similar to a symmetry constraints or the fluctuationless requirement of classical population models discussed above. The formalism discussed here can be adapted to more general theories with different or additional dark states by modifying the nature of this constraint.

The partition function for the Keldysh field theory is:

$$Z = \int D\phi^\alpha D\bar{\phi}^\alpha \, e^{iS[\phi^c, \phi^q]}, \tag{30a}$$

$$S[\phi^c, \phi^q] = \int dx \left( \bar{\phi}^q i\partial_t \phi^c + \bar{\phi}^c i\partial_t \phi^q - \mathcal{K}(\phi^\alpha, \bar{\phi}^\alpha) \right), \tag{30b}$$

where $\phi^\alpha$ the Keldysh basis of fields defined by the standard rotation $\phi^{c,q} = (\phi^+ \pm \phi^-)/\sqrt{2}$, with $\phi^\pm$ the field variables corresponding to the bosonic operator $\hat{a}$ on the forward/backward part of the Keldysh time contour, $\phi^\pm \leftrightarrow \hat{a}_\pm$, $\bar{\phi}^\pm \leftrightarrow \hat{a}^\dagger_\pm$, [52]. Performing such substitutions in the Lindbladian superoperator, Eqs. (9), (10), one obtains the Keldysh "Hamiltonian", $\mathcal{K}(\phi^c, \bar{\phi}^c, \phi^q, \bar{\phi}^q)$. Generically it includes any terms allowed from either the Hamiltonian and dissipative parts of the Lindbladian dynamics [15, 17]. The $N$-point functions, as in Eq. (11), are given by path integral expectations,

$$\left\langle \phi^{\sigma_N}_{j_N}(t_N) \cdots \bar{\phi}^{\sigma_2}_{j_2}(t_2) \phi^{\sigma_1}_{j_1}(t_1) \right\rangle_\rho = \langle\!\langle 1 | \hat{a}_{j_N \sigma_N} \hat{\mathcal{U}}_{t_N - t_{N-1}} \cdots \hat{a}^\dagger_{j_2 \sigma_2} \hat{\mathcal{U}}_{t_2 - t_1} \hat{a}_{j_1 \sigma_1} \hat{\mathcal{U}}_{t_1} | \rho \rangle\!\rangle, \tag{31}$$

where $\sigma \in \{+, -\}$.

To conserve probability, the Keldysh action must obey $S[\phi^c, \phi^q = 0] = 0$. Similar to the field theory for classical population models the Keldysh action must also obey a parallel property for the classical fields

$$S[\phi^c = -\phi^q, \bar{\phi}^c = \bar{\phi}^q] = 0. \tag{32}$$

This ensures the action vanishes when $\phi^+ = 0 = \bar{\phi}^-$, which corresponds to the action of $\hat{a}$ on the left and $\hat{a}^\dagger$ on the right of the density matrix, both of which vanish in the extinct dark state, $|0\rangle$. This constraint on the action comes from the fluctuationlessness of observables and charged operators, as per by Eq. (19). For example, any dark state expectations of either $\phi^\alpha$ (without $\bar{\phi}^\alpha$) or the number $n^\pm = \bar{\phi}^\pm \phi^\pm$ is identically zero,

$$\left\langle \phi^{\alpha_N}(x_N) \cdots \phi^{\alpha_2}(x_2) \phi^{\alpha_1}(x_1) \right\rangle = 0, \tag{33a}$$

$$\left\langle n^{\alpha_N}(x_N) \cdots n^{\alpha_2}(x_2) n^{\alpha_1}(x_1) \right\rangle = 0, \tag{33b}$$

with $n^{c,q} = (n^+ \pm n^-)/2$. Unlike the classical theory discussed in the previous section in which all the expectations of only the classical number field $n$ vanish in the absorbing state Eq. (27), certain dark state expectations of the classical fields $\phi^c$, $\bar{\phi}^c$ do *not* vanish. This is reminiscent of differences between classical and quantum models at zero temperature: quantum systems will have vacuum fluctuations even in the absence of thermal excitations while classical systems are totally inert. Correlation functions of only classical fields are nevertheless constrained by the dark state condition, being expressible in terms of other mixed expectations involving $\phi^q$, $\bar{\phi}^q$.

We make this idea precise at the level of two-point function. The spectral and Keldysh components of the Green's functions are:

$$\begin{bmatrix} iG^{\mathrm{K}}(x,x') & iG^{\mathrm{R}}(x,x') \\ iG^{\mathrm{A}}(x,x') & 0 \end{bmatrix} = \left\langle \begin{bmatrix} \phi^{\mathrm{c}}(x) \\ \phi^{\mathrm{q}}(x) \end{bmatrix} \begin{bmatrix} \bar{\phi}^{\mathrm{c}}(x') & \bar{\phi}^{\mathrm{q}}(x') \end{bmatrix} \right\rangle. \tag{34}$$

The Keldysh and spectral components are related by the distribution function $F(x,x')$ by $G^{\mathrm{K}} = G^{\mathrm{R}} \circ F - F \circ G^{\mathrm{A}}$ where $\circ$ denotes operator composition of spacetime arguments. In the Keldysh theory for non-interacting Lindbladians $F$ is always a delta-correlated in the stationary state, with $F_{\mathrm{st}} = \delta(x,x')$ for $\rho_{\mathrm{st}} = |0\rangle\langle 0|$ [17]. For a quantum population model this property is *exact* even when interactions are present.[3] Note also that the Nambu components of any Green's functions are automatically zero in the dark state as a consequence of Eq. (33). Thus in total one finds for $|0\rangle$ a dark state:

$$G^{\mathrm{K}} = G^{\mathrm{R}} - G^{\mathrm{A}}, \tag{35a}$$

$$\langle \phi^{\alpha} \phi^{\beta} \rangle = 0 = \langle \bar{\phi}^{\alpha} \bar{\phi}^{\beta} \rangle. \tag{35b}$$

In other words *there is an exact relationship between Keldysh and spectral Greens function that holds beyond perturbation theory.* Similar conditions will appear in any theory with a dark state, though their exact form will depend on the details of the dark state(s) present.

This situation is somewhat similar to the KMS condition in the equilibrium Keldysh theory, which relates retarded and advanced correlation functions to Keldysh correlation functions via the FDT and its generalizations. This fact in the equilibrium theory allows for the development of the Matsubara formalism, which dramatically simplifies many calculations by recasting everything in terms of a single Green's function, the imaginary time Matsubara function. In the present non-equilibrium context, the dark state condition makes a similar reduction possible. The formal tool to achieve this is known as the "third quantized" basis of superoperators [59–63] or its Keldysh field theory equivalent [17, 64]. This method was originally developed to study non-interacting Lindbladian theories and relies on the fact that $F$ is delta-correlated in time. For general interacting Lindbladians which generically may have off-diagonal in time $F$, such a reduction is not feasible. However, for models with dark states, where $F$ is required to be delta-correlated in time, it can naturally be extended to the many-body setting. This approach allows for the evaluation of stationary expectations to be carried out using a single *real-time* Green's function. This considerably simplifies certain calculations, most notably for our purposes perturbative RG. We note that this formalism does not provide an advantage when considering non-stationary distributions, as then $F$ is time dependent and not generically delta-correlated and thus the third-quantized basis becomes ill-defined. If one is interested in such quantities, more standard approaches based on Keldysh kinetic theory would likely be more suitable; some progress has been made to develop a Keldysh kinetic description of quantum population models in [28].

---

[3]Consider $t > t'$. In the superoperator language, $G^{\mathrm{K}}(t,t') = \langle\!\langle 1 | \hat{a}_c \hat{\mathcal{U}}_{t-t'} \hat{a}_c^{\dagger} | \rho \rangle\!\rangle$ and $G^{\mathrm{R}}(t,t') = \langle\!\langle 1 | \hat{a}_c \hat{\mathcal{U}}_{t-t'} \hat{a}_q^{\dagger} | \rho \rangle\!\rangle$. In the the extinct stationary state $\rho = |0\rangle\langle 0|$, one has $\hat{a}_c^{\dagger} | \rho \rangle\!\rangle = \frac{1}{\sqrt{2}} \hat{a}_+^{\dagger} | \rho \rangle\!\rangle = \hat{a}_q^{\dagger} | \rho \rangle\!\rangle$ and thus $G^{\mathrm{K}} = G^{\mathrm{R}}$. A similar argument shows $G^{\mathrm{R}} = -G^{\mathrm{A}}$ when $t < t'$, which together gives the first equation in Eq. (35). For the second equation, for $t > t'$ one has $\langle \phi^{\alpha}(t) \phi^{\beta}(t') \rangle = \langle\!\langle 1 | \hat{a}_{\alpha} \hat{\mathcal{U}}_{t-t'} \hat{a}_{\beta} | \rho \rangle\!\rangle \propto \langle\!\langle 1 | \hat{a}_{\alpha} \hat{\mathcal{U}}_{t-t'} | \rho \hat{a} \rangle\!\rangle$. Because of the dark state condition, the dynamics only evolve the right side of the operator $\rho \hat{a}$. Indeed, so long as $|0\rangle$ is a dark state, $\hat{\mathcal{L}}(\rho \hat{a}) = \rho \hat{a}(i\hat{H} - \frac{1}{2}\sum_{\nu} \gamma_{\nu} \hat{L}_{\nu}^{\dagger} \hat{L}_{\nu})$. As a consequence, one has $\hat{\mathcal{U}}_{t-t'}(\rho \hat{a}) = \sum_n c_n |0\rangle\langle n|$ where the specific coefficients $c_n$ depend on the details of the dynamics. Thus, $\langle\!\langle 1 | \hat{a}_{\alpha} \hat{\mathcal{U}}_{t-t'} \hat{a}_{\beta} | \rho \rangle\!\rangle = \sum_n c_n \mathrm{tr}(\hat{a}_{\alpha} |0\rangle\langle n|)$. For $\hat{a}_{\alpha} = \hat{a}_{c,q}$ this is $\propto \sum_n c_n \mathrm{tr}(\hat{a} |0\rangle\langle n| \pm |0\rangle\langle n| \hat{a})$, which is identically zero because the first term here is annihilated by $\hat{a} |0\rangle = 0$ and the second term is a traceless operator inside the trace. This gives the second equation in Eq. (35), and moreover a generalization of this argument also demonstrates the validity of Eq. (33a).

$$\mathcal{G}(x,x') \qquad\qquad \bar{\mathcal{G}}(x,x')$$

Figure 1: Diagrammatic representation of the dark space Green's function $\mathcal{G}$. Solid and dashed lines represent the classical and auxiliary fields $\phi$ and $\chi$; in- and out-going lines denote fields and their conjugates.

This representation is obtained by diagonalizing the quadratic part of the Keldysh action. For the quantum population dynamics, the most generic quadratic action[4] has the structure:

$$S_0 = \int dx \begin{bmatrix} \bar{\phi}^c & \bar{\phi}^q \end{bmatrix} \begin{bmatrix} 0 & i\partial_t - \Delta_{\mathbf{r}} - i\sum_{\nu}\bar{\mu}_{\nu\mathbf{r}}\mu_{\nu\mathbf{r}} \\ i\partial_t - \Delta_{\mathbf{r}} + i\sum_{\nu}\bar{\mu}_{\nu\mathbf{r}}\mu_{\nu\mathbf{r}} & 2i\sum_{\nu}\bar{\mu}_{\nu\mathbf{r}}\mu_{\nu\mathbf{r}} \end{bmatrix} \begin{bmatrix} \phi^c \\ \phi^q \end{bmatrix}, \tag{36}$$

where the Hamiltonian and dissipative contributions, $\Delta_{\mathbf{r}}$ and $\mu_{\nu\mathbf{r}}$ correspondingly, may include both constants or spatially local differential operators. Diagonalizing this quadratic form is achieved by introducing the new classical and quantum fields:

$$\phi = \phi^c + \phi^q, \qquad \bar{\phi} = \bar{\phi}^c - \bar{\phi}^q, \qquad \chi = -\phi^q, \qquad \bar{\chi} = \bar{\phi}^q. \tag{37}$$

Naively in this new basis $\bar{\phi}$ and $\bar{\chi}$ are not the complex conjugates of $\phi$ and $\chi$. This problem can be fixed by deforming the original functional integration contour, in particular by complexifying the space of only the quantum fields so that $\bar{\chi} = \chi^*$ is the complex conjugate of $\chi$, from which it follows also that $\bar{\phi} = \phi^*$. Letting $iS[\phi^\alpha] = -S[\phi,\chi]$, the resulting action including both the quadratic part $S_0$ and any interactions is:

$$S[\phi,\chi] = \int dx \left( \bar{\chi}\partial_t\phi + \chi\partial_t\bar{\phi} - K(\phi,\bar{\phi},\chi,\bar{\chi}) \right), \tag{38a}$$

$$K(\phi,\bar{\phi},\chi,\bar{\chi}) = \bar{\chi}\left(\Delta_{\mathbf{r}} - i\sum_{\nu}\bar{\mu}_{\nu\mathbf{r}}\mu_{\nu\mathbf{r}}\right)\phi + \ldots + \text{c.c.}, \tag{38b}$$

where $(\ldots)$ indicates any nonlinear terms included in the theory. The effective Hamiltonian $K(\phi,\bar{\phi},\chi,\bar{\chi})$ is always real and thus defines an invariant subspace of the motion of the full Keldysh quasi-classical mechanics. The two-point functions define a single new Green's function,

$$\mathcal{G}(x,x') = \langle\phi(x)\bar{\chi}(x')\rangle, \qquad \bar{\mathcal{G}}(x,x') = \langle\chi(x)\bar{\phi}(x')\rangle, \tag{39a}$$

$$\langle\phi(x)\chi(x')\rangle = 0 = \langle\bar{\phi}(x)\bar{\chi}(x')\rangle, \tag{39b}$$

$$\langle\phi(x)\bar{\phi}(x')\rangle = 0 = \langle\phi(x)\phi(x')\rangle, \qquad \langle\chi(x)\bar{\chi}(x')\rangle = 0 = \langle\chi(x)\chi(x')\rangle, \tag{39c}$$

which follows immediately from expressing Eq. (35) in terms of the new fields $\phi$ and $\chi$. In particular, $\langle\phi(x)\bar{\phi}(x')\rangle = G^K - G^R + G^R = 0$. Noting also that $\langle\phi\bar{\chi}\rangle = \langle\phi^c\bar{\phi}^q + \phi^q\bar{\phi}^q\rangle$, the functional form of the diagonal Green function $\mathcal{G}$ is always related to the standard retarded Green's function $\mathcal{G}(x,x') = iG^R(x,x')$. We choose the convention for $\bar{\mathcal{G}}$ so that in Fourier space $\bar{\mathcal{G}}(p) = (\mathcal{G}(p))^*$. The diagrammatic representation of these Green's functions is depicted in Fig. 1.

The action (38), resulting from this procedure, always respects the conditions of probability conservation and fluctuationlessness of the dark state,

$$S[\phi,\chi=0] = 0 = S[\phi=0,\chi]. \tag{40}$$

---

[4]This is the most generic form of action for a quadratic Lindbladian with spatial locality for which the Fock vacuum is a stationary state [17]. Such an action corresponds to the linear Hamiltonian $\hat{H} = \int_{\mathbf{r}} \hat{a}_{\mathbf{r}}^\dagger \Delta_{\mathbf{r}} \hat{a}_{\mathbf{r}}$ and jump operators $\hat{L}_{\nu\mathbf{r}} = \mu_{\nu\mathbf{r}}\hat{a}_{\mathbf{r}}$.

This is the analog of the condition (26), respected by the classical population models. Thus, similar to the classical theory, quantum population models are characterized by the conditions placed on the action by the dynamical constraints, imposed by the existence of the dark state. The primary conceptual difference is that the quantum theory is defined on the level of the bosonic field $\phi \sim \sqrt{n}$. It is this fact that allows the inclusion of superpositions of classical reaction processes, such as the cat process, Eq. (20). In the quantum theory the population number is associated with a two-point correlation function: recalling that $\phi = \sqrt{2}\phi^+$ and $\bar{\phi} = \sqrt{2}\bar{\phi}^-$, one has $\langle \hat{n}(\mathbf{r}) \rangle = \langle \bar{\phi}\phi \rangle /2$. This has significant consequences on the critical scaling of the population near the dark state transition, as will be discussed below.

## 4 Universal theory of Schrödinger cat populations

We apply the above formalism to develop a universal field theory for a quantum population dynamics of a single species with no additional symmetries or constraints beyond locality and spacial homogeneity. We find two phases: an active phase, which hosts a quantum superposition of finite stable populations, and an extinct "dark" phase. By tuning a single complex parameter these two phases are connected by a continuous transition. The RG reveals the transition to belong to a distinct universality class, though some (but not all) of its critical indexes coincide with the DP class (at least to the one loop order, considered here).

In the absence of additional symmetries, there are three sources of quantum features that end up being relevant in the RG sense: cat processes and superpositions of death processes $\hat{L}_c \sim \hat{a}^\dagger \hat{a} + \hat{a}$ and $\hat{a}^2 + \hat{a}$, incoherent quantum hopping $\hat{L} \sim \partial_\mathbf{r}\hat{a}$, and Hamiltonian terms. The incoherent processes alone describe the dynamics of a population of animals which can both multiply and die in the standard classical way and also enter superpositions of being simultaneously alive and dead or being in different locations in space. In models with only classical population processes the Hamiltonian plays no role because dynamics are restricted to the diagonal of the charge basis. When quantum population processes are present this restriction is lifted and dynamics spread into the entire Fock space, so coherent effects also affect the dynamics.

Microscopically the Hamiltonian can contain any operators of the schematic form $\hat{a}^\dagger \hat{O} \hat{a}$ and the jump operators must take the general form $\hat{L}_v = \hat{O}_v \hat{a}$. We place no restrictions on the operators $\hat{O}$, $\hat{O}_v$ besides spatial homogeneity and locality. The specific form of allowed jump operators and Hamiltonians is discussed in Appendix A. A resulting universal action for such models is obtained by expanding the generic action of Eq. (38) to lowest non-trivial terms in both $\phi(x)$ and $\chi(x)$ complex fields,

$$S[\phi,\chi] = \int dx \left( \bar{\chi}(\partial_t - D\partial_\mathbf{r}^2)\phi - \bar{\chi}(\alpha + \beta_1 \phi + \beta_2 \bar{\chi} + \beta_3 \bar{\phi})\phi \right) + \text{c.c.}, \qquad (41)$$

where $\alpha$, $\beta_j$, and $D$ are all complex parameters (see Appendix A for their relation to microscopic reaction rates). Their real parts are set by the strength of the quantum population processes, corresponding respectively to the rates of classical death, cat processes, and incoherent quantum hopping.[5] The imaginary parts of the parameters come from Hamiltonian contributions.

The action of Eq. (41) is a universal action for the quantum Schrödinger cat population dynamics. It may be obtained without appeal to specific details of microscopic reaction rules by

---

[5]Naively it seems that only a negative real part for $\alpha$ is permitted, as this corresponds to a positive rate for the death process, $\hat{L}_d = \hat{a}$. Moreover, there is no microscopic operator one can construct from $\hat{a}$ and $\hat{a}^\dagger$ that encodes the linear reproduction processes $A \to 2A$ by itself. However, higher-order branching/reproduction processes (for example, $\hat{L} = \hat{a}^{\dagger 2}\hat{a}$, see table 1) can renormalize the real part of the mass, $\alpha$, to a positive value.

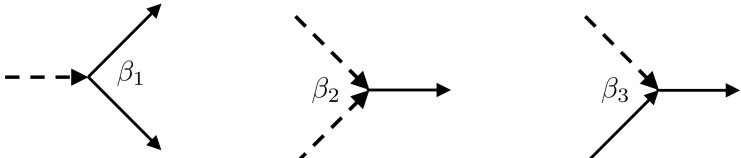

Figure 2: Allowed vertices of the Schrödinger cat population field theory. The conjugate vertices with coupling $\bar{\beta}_j$ are also included and are given by reversing the directions of all arrows.

writing all RG relevant terms allowed by the conditions imposed on the action and correlation functions by the existence of the dark state as encoded by Eqs. (39) and (40). Such constraints are not readily reducible to a symmetry, but are nevertheless preserved under RG and restrict the allowed vertices in the action. The three cubic vertices, given above and for which the corresponding diagrams are shown in Fig. 2, are the only allowed vertices with three fields that do not violate these conditions.[6] The microscopic origin of the cubic vertices can be traced (see Appendix A) to both the cubic terms in the Hamiltonian and the quantum reaction process $\hat{L}_c \sim \hat{a}^\dagger \hat{a} + \hat{a}$ and $\hat{a}^2 + \hat{a}$ (in particular the cross terms of the dissipative superoperator containing three powers of $\hat{a}$ and $\hat{a}^\dagger$). Such terms have the schematic form $\sim \hat{a}^\dagger \hat{a} \hat{a}$. They correspond to both coherent and incoherent processes that change the particle number on one side of the density matrix, thus producing superpositions of different number states.

Any vertices with higher than the third power in the fields are less relevant in the RG sense. Indeed, from the linear part of the action one finds that bare value of scaling dimensions $\boldsymbol{\Delta}_\phi$ and $\boldsymbol{\Delta}_\chi$ of $\phi$ and $\chi$ are $\boldsymbol{\Delta}_\phi = -d/2 = \boldsymbol{\Delta}_\chi$. When accounting for fluctuations, both $\boldsymbol{\Delta}_\phi$ and $\boldsymbol{\Delta}_\chi$ receive perturbative corrections, however the relation $\boldsymbol{\Delta}_\phi = \boldsymbol{\Delta}_\chi$ remains intact. The equality of the classical and auxiliary fields' scaling dimensions is a property shared with the classical directed percolation field theory. This is also matches to what is found for "quantum" scaling in other Lindblad models [16, 56]. Noting also the bare value for the dynamical exponent $z = 2$, one sees that local third order terms are marginal for $d = 4$ and are the only relevant vertices for $4 > d \geq 3$.

Despite the formal similarity to the action for classical directed percolation, Eq. (23), the Schrödinger cat population action (41) has a very different physical meaning. Eq. (23) cannot be retrieved as a limit of Eq. (41). The primary fields $\phi$ and $\chi$ are complex variables and may take values in the entire complex plane. The fluctuationless equation of motion for the classical field resembles the classical noiseless FKPP equation Eq. (25), but with a complex variable:

$$(\partial_t - D\partial_{\mathbf{r}}^2)\phi = (\alpha + \beta_1\phi + \beta_3\bar{\phi})\phi \,. \tag{42}$$

In addition to $\phi = 0$, this equation has another spatially homogeneous solution:

$$\underline{\phi} = \frac{\bar{\alpha}\beta_3 - \alpha\bar{\beta}_1}{\bar{\beta}_1\beta_1 - \bar{\beta}_3\beta_3} \,. \tag{43}$$

In a dark phase, the $\phi = 0$ solution must be dynamically stable and other fixed points must be unstable. An active phase alternatively must have an unstable dark state and a stable finite population, Eq. (43). If both fixed points are fail to be dynamically stable, the model requires higher-order nonlinearities to stabilize and thus depends on the details of higher-order vertices.

---

[6]Naively it appears that additional terms like $\bar{\chi}\chi\phi$, $\chi\phi\phi$, etc. may be allowed, as they respect Eq. (40). This is however not the case: such terms never appear from a microscopic construction, as discussed in Appendix A. Moreover, their naive inclusion in the effective field theory leads to perturbative violations of Eq. (39). As an example, note that the inclusion of a vertex $\chi\phi\phi$ permits contraction with the $\beta_2$ vertex: $\langle(\chi\phi\phi)(\bar{\chi}\bar{\chi}\phi)\rangle \sim \langle\phi\bar{\chi}\rangle\langle\phi\bar{\chi}\rangle\chi\phi$ and thus generates a forbidden $\chi\phi$ coupling.

Such situations generically correspond to first-order transitions and will not be considered here in detail.

A phase transition separating a dark phase and an active phase can be found in any dimension $d > 0$. When the transition is continuous, it can be studied using standard perturbative RG by expanding near the critical dimension $d_c = 4$ in $\epsilon = 4 - d$, details of the critical scaling in $4 > d \geq 3$ are presented in subsequent sections. For $d \geq 4$, all non-linear vertices have a RG irrelevant scaling and so the critical scaling becomes mean-field (though one may infer that fluctuations may still have small effects like renormalizing the critical value of $\alpha$, as is known to occur in classical reaction models [74,75]). For $d < 3$, quartic and other higher-order vertices may become relevant but the cubic vertices will remain the most relevant interactions. However the $\epsilon$ expansion will become uncontrolled and perturbative RG will no longer provide a reliable estimate of the critical exponents. Despite this, just as in classical directed percolation the phase transition should be expected to persist in low dimensions. Before addressing the full model, we first consider several specific limiting cases.

## 4.1 Extinction of Schrödinger cats in zero dimensions

As a warm up, we first consider the Schrödinger cat dynamics in zero spatial dimensions. This is described by the dissipative quantum mechanics of a single bosonic mode. For generic parameter values there are no symmetries and so the dark state with zero population is the unique stationary state. The inclusion of higher order processes may destabilize the dynamics and lead to the uncontrolled growth of population. Much like in classical population dynamics, in zero dimensions these are generically the only two possibilities: all populations eventually go extinct or diverge to infinity at long times.

To demonstrate this, we will study a simplified example with no Hamiltonian and only the cat process $\hat{L}_c \sim \hat{a}^\dagger \hat{a} + \hat{a}$ and classical birth and death $A \rightarrow \emptyset$ and $A \rightarrow 2A$. This is equivalent to having real $\alpha, \beta_1, \beta_2$, with $\beta_1 = \beta = \beta_2/2$, and $\beta_3 = 0$. The effective Hamiltonian is then:

$$K = \bar{\chi}(\alpha + \beta\phi + 2\beta\bar{\chi})\phi + \text{c.c.} \tag{44}$$

We will examine the fate of initially small populations and focus on the regime of small $\beta$. To do so, consider the full equations of motion of the Keldysh semiclassical dynamics,

$$\partial_t \phi = (\alpha + \beta\phi + 4\beta\bar{\chi})\phi, \tag{45a}$$

$$\partial_t \bar{\chi} = -\bar{\chi}(\alpha + 2\beta\bar{\chi} + 2\beta\phi). \tag{45b}$$

The classical dynamics of $\phi$ is described by the motion in the $\chi = 0$ subspace. The nature of the dynamics depends on the sign of $\alpha$. For $\alpha < 0$, $\phi = 0$ is a stable fixed point of the motion. There is an additional classical fixed point at $\phi = -\alpha/\beta$, but it is unstable. At sufficiently late times, populations decay exponentially with the rate set by $\alpha$.

There is a bifurcation of the dynamics when $\alpha = 0$ caused by the merging of the two classical fixed points at $\phi = 0$. For $\alpha > 0$ the stability of the two fixed points flips: the origin becomes an unstable source and the $\phi = -\alpha/\beta$ fixed point becomes classically stable. Incorporating the effects of fluctuations generically shows this state to be only meta-stable. This is due to rare extinction events in which populations near the meta-stable value undergoes an activated transition to zero population. Activation is generally associated with instanton trajectories of the semiclassical motion that connect active fixed points to $\phi = 0$ along which the auxiliary field $\chi$ is finite. The result is a finite, though exponentially small, extinction time. This feature is shared by both classical and dissipative quantum systems.

The semiclassical dynamics of the cat processes is simple enough that such an activation trajectory can be identified. To achieve this, note that $\phi$ and $\chi$, both real, define a two-dimensional invariant subspace of the motion. This subspace contains both of the classical

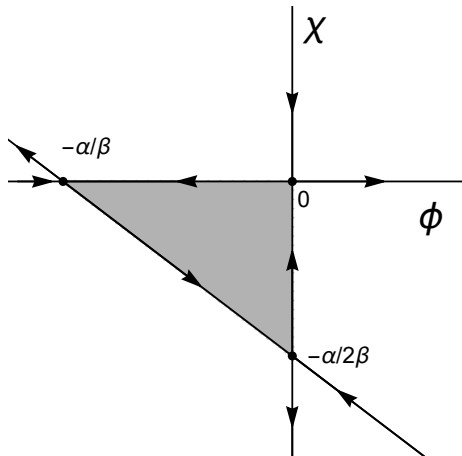

Figure 3: Real slice of the phase portrait of the Keldysh classical dynamics governed by Eq. (45) for $\alpha > 0$. The area of gray highlighted region is equal to the action, $S = \int \chi \, d\phi$, accrued along the activation trajectory and thus determines the asymptotic extinction time.

fixed points and an additional quantum fixed point with $\phi = 0$ and $\chi = -\alpha/2\beta$. There are three lines with $K = 0$ connecting these three fixed points, defined by the conditions $\phi = 0$, $\chi = 0$, and $\chi = -(\phi + \alpha/\beta)/2$. The phase portrait thus obtains a triangular topology, as depicted in Fig. 3. This situation directly parallels minimal classical population models [65, 76], the difference being that in the present context the nonlinearity comes from the cat process instead of higher-order classical population processes. The activation trajectory goes diagonally from the active fixed point, $\phi = -\alpha/\beta$, $\chi = 0$, to the quantum fixed point, $\phi = 0$, $\chi = -\alpha/2\beta$ and then to the dark state, $\phi = 0 = \chi$, along the $\chi$ axis. The action accumulated along this trajectory is $S = \alpha^2/4\beta^2$. A saddle point approximation thus predicts a long extinction time $\propto \exp(\alpha^2/4\beta^2)$, which holds in the limit of weak nonlinearity $\beta < \alpha$.

## 4.2  Complex Reggeon theory

We now consider the previous example in the full finite-dimensional setting. In terms of microscopic reaction rules (see Appendix A), $\beta_3 = 0$ when there are no cubic terms in the Hamiltonian and the only incoherent processes are cat processes $\hat{L}_c \sim \hat{a}^\dagger \hat{a} + \hat{a}$ and classical birth and death $A \to \emptyset$ and $A \to 2A$. With $\beta_3 = 0$ the theory has a complex version of the Reggeon inversion symmetry, Eq. (28), displayed by the classical population dynamics,

$$\phi(t, \mathbf{r}) \to \frac{\beta_2}{\beta_1} \bar{\chi}(-t, \mathbf{r}), \qquad \bar{\chi}(t, \mathbf{r}) \to \frac{\beta_1}{\beta_2} \phi(-t, \mathbf{r}). \tag{46}$$

A finite value for $\beta_3$ breaks this symmetry, and so a $\beta_3$ term is not generated by RG, if it is not present microscopically.

The fluctuationless equation of motion is:

$$(\partial_t - D\partial_{\mathbf{r}}^2)\phi = (\alpha + \beta_1\phi)\phi. \tag{47}$$

The two stationary homogeneous solutions are zero and $\underline{\phi} = -\alpha/\beta_1$. The stability of both of these solutions depends on the real part of the complex mass, $\alpha$. For $\mathrm{Re}(\alpha) < 0$ the $\phi = 0$ is a stable spiral sink and the $\underline{\phi}$ fixed point is an unstable spiral source. This is a dark phase, in which the extinct state is dynamically stable and the stationary state population is zero. The situation is reversed for $\mathrm{Re}(\alpha) > 0$. Now, the $\phi = 0$ fixed point becomes unstable and the $\underline{\phi}$

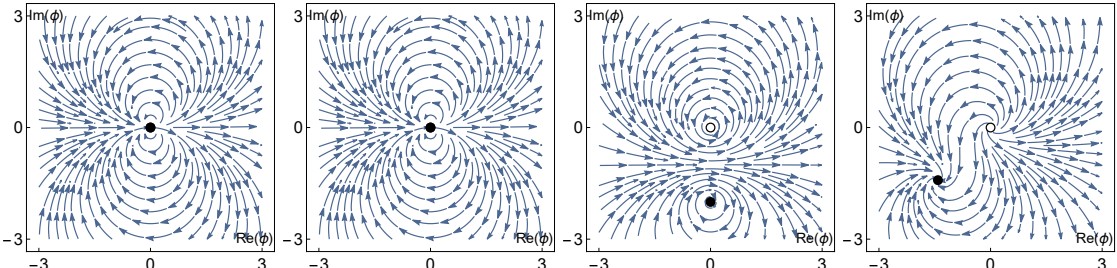

Figure 4: Plots of various trajectories of the mean-field dynamics of Eq. (47) demonstrating the stability of the fixed points for different dynamical regimes. In all plots, the horizontal and vertical axes correspond to the real and imaginary parts of $\phi$ and the white and black points show the dark and active fixed points, correspondingly. All plots are made with $|\alpha| = 2$ and $\beta_1 = 1$. The leftmost plot shows the dead phase with $\arg(\phi) = 3\pi/4$. The second to the left shows the critical state, $\alpha = 0$. The second from the right shows two limit cycles, with $\arg(\alpha) = \pi/2$. The rightmost shows the active phase with $\arg(\alpha) = \pi/4$.

fixed point becomes stable, thus defining an active phase with a finite stationary population $\langle n \rangle = \bar{\alpha}\alpha/2\bar{\beta}_1\beta_1$. Fig. 4 shows the mean-field dynamics in the complex $\phi$-plane for both phases.

The stability of the two solutions changes discontinuously when the $\mathrm{Re}(\alpha)$ changes sign at a finite $\mathrm{Im}(\alpha)$. Putting $\mathrm{Re}(\alpha) = 0$ with finite $\mathrm{Im}(\alpha)$ results in a limit cycle and the $\phi = 0$ fixed point becomes purely oscillatory. This leads to a jump in the the stationary state population as the real $\alpha$ axis is crossed, thus indicating a first order transition between the active and dark phases. A continuous transition requires the simultaneous vanishing of both components of the complex mass, $\alpha$. At the mean-field level, this corresponds to coalescing of the two fixed points at $\phi = 0$ into a single double-degenerate fixed point (Fig. 4 second panel). A mean field phase diagram is shown in the first panel of Fig. 5; while fluctuations may renormalize the value of the critical value of $\alpha$ this picture is expected to hold qualitatively.

The nature of the critical fluctuations turns out to be reminiscent of those in the classical directed percolation universality class. Despite this, there are key differences between the classical and quantum models. This can be appreciated even on the mean-field level. As an example, from the above discussion one finds the population as a function of the classical death rate near critical point scales at $\langle n \rangle \sim |\alpha|^2$. This contrasts with the classical population dynamics where $\langle n \rangle \sim |\alpha|^1$ at the mean-field level, see Eq. (29). Similarly, at the critical point, $\alpha = 0$, the mean-field predicts long time behavior of the population density $\langle n(t) \rangle \sim 1/t^2$, as opposed to $1/t$ behavior in the classical models. The difference is due to the different symmetry class of the complex Reggeon theory, which may be traced back to the breaking of the local weak $U(1)$ symmetry by quantum reaction rules.

We investigate the nature of the critical fluctuations in detail using a standard perturbative RG approach [77,78]. This is achieved by means of an $\epsilon$-expansion around the critical dimension $d_c$, for which we find $d_c = 4$. The results of this calculation are summarized here and details of the calculation are discussed in Appendix B. The coefficient $Z$ will be used for the kinetic term $\bar{\chi}\partial_t\phi$, which has a bare value of $Z = 1$. To lowest order in loops there is a single diagram that contributes to each parameter in the action; they are depicted in Fig. 6.

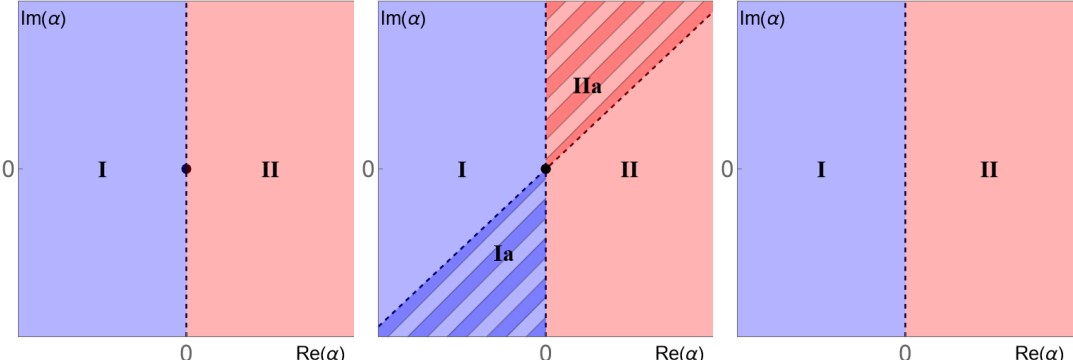

Figure 5: Phase diagrams for the mean-field dynamics of the quantum population dynamics. The first panel shows the complex Regge theory, in which $\beta_3 = 0$. The middle panel shows $0 < |\beta_3| < |\beta_1|$ (in particular, $|\beta_3| = |\beta_1|/2$ and $\arg(\beta_3) + \arg(\beta_1) = 3\pi/8$ is shown); the right panel shows $|\beta_3| > |\beta_1|$. In all panels, phase I (colored in blue) is the dark phase and phase II (colored in red) is the active phase. The dark circle denotes the origin; a continuous transition is obtained only by crossing this point. On dashed lines the non-zero fixed point $\phi$ is a limit cycle; crossing the these lines results in a first order transition in which $\overline{\langle \phi \rangle}$ changes discontinuously. In the middle panel, phases Ia and IIa label the values of $\alpha$ for which the fixed points are both sinks/sources resp.

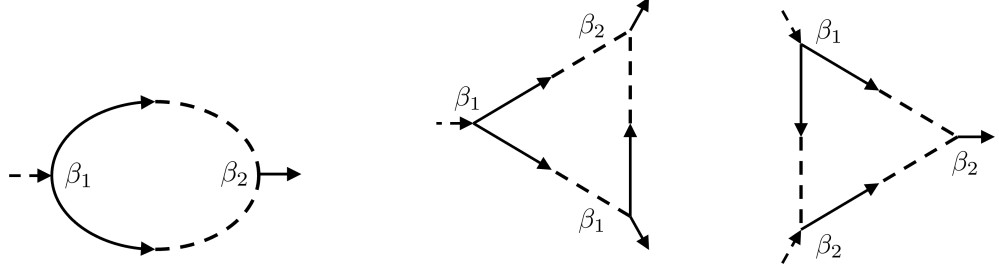

Figure 6: 1-loop diagrams contributing to the renormalization of the propagator and the two vertices $\beta_1$ and $\beta_2$. As, roughly speaking, in/outgoing arrows correspond to annihilation and creation of particles, each diagram can be thought of intuitively as a sequence of branching and recombination processes. Such sequences occur on either side of the density matrix independently and thus produce superpositions of different particle numbers.

They result in the ten RG flow equations:

$$\partial_l Z = \left(d + \Delta_\phi + \bar{\Delta}_\chi + \frac{C_d \beta_1 \beta_2}{2ZD^2}\right)Z\,, \quad \partial_l D = \left(d + z - 2 + \Delta_\phi + \bar{\Delta}_\chi + \frac{C_d \beta_1 \beta_2}{4ZD^2}\right)D\,, \quad \text{(48a)}$$

$$\partial_l \alpha = \left(d + z + \Delta_\phi + \bar{\Delta}_\chi + \frac{C_d \beta_1 \beta_2}{ZD^2}\right)\alpha\,, \quad \text{(48b)}$$

$$\partial_l \beta_1 = \left(d + z + 2\Delta_\phi + \bar{\Delta}_\chi + \frac{2C_d \beta_1 \beta_2}{ZD^2}\right)\beta_1\,, \quad \partial_l \beta_2 = \left(d + z + \Delta_\phi + 2\bar{\Delta}_\chi + \frac{2C_d \beta_1 \beta_2}{ZD^2}\right)\beta_2\,, \quad \text{(48c)}$$

and their complex conjugates, where $l$ is the flowing scale and $C_d$ is a real constant that depends on the both the UV cutoff and the spatial dimension (see Appendix B). These are very similar to 1-loop flow equations for standard real Regge field theory [46, 49, 52], but have several important differences which we subsequently discuss.

To identify a fixed point, it is convenient to first observe that four of the above equations can be wrapped up into a single, self-contained flow equation for the parameter $U = \beta_1 \beta_2 / Z D^2$,

$$\partial_l U = (\epsilon + 3C_d U)U, \tag{49}$$

with $\epsilon = 4 - d$. One thus observes a non-trivial fixed point with $U^\star = -\epsilon/3C_d$ and $\alpha^\star = 0$. Feeding this back into Eq. (48), one finds $\Delta_\phi = -2 + 7\epsilon/12 = -d/2 + \epsilon/12$, $\Delta_\chi = \Delta_\phi$ (which is anyways demanded by the Reggeon inversion symmetry), and $z = 2 - \epsilon/12$. The values of $Z, D$, and $\beta_{1,2}$ are not uniquely specified by this procedure, thus leading to the conclusion that there is in fact a critical *surface* of complex dimension 2: the ratio $\beta_1/\beta_2$ is fixed prima facie and does not flow under RG because of its participation in Eq. (46) and the condition $U = U^\star$ fixes only one of the remaining three couplings.

All points on this critical surface turn out to be stable in the direction of the nonlinear couplings and unstable in the $\alpha$ direction (see Appendix C). The critical scaling is the same on the entire surface, with the critical exponents,

$$z = 2 - \frac{\epsilon}{12}, \qquad \nu = \frac{1}{2} + \frac{\epsilon}{16}, \qquad \alpha = 1 - \frac{\epsilon}{4}, \qquad \beta = 1 - \frac{\epsilon}{6}, \tag{50}$$

which match the directed percolation exponents from Eq. (29) up to the one-loop level. For details on how the exponents are obtained, see Appendix D.

Note however that in the present context the order parameter exponents $\alpha$ and $\beta$ are defined by the critical scaling of $\phi$ and not the population number, with $\langle \phi \rangle \sim t^{-\alpha}$ and $\langle \phi \rangle \sim |\alpha|^\beta$. The critical scaling of the population number in the quantum model defines a new exponent, owing to the fact that $n = \bar\phi \phi/2$ is a composite operator in the RG sense in the quantum theory. Thus, we introduce the exponents $\langle n \rangle \sim |\alpha|^{\beta_2}$ and $\langle n(t) \rangle \sim t^{-\alpha_2}$. At order $\epsilon$, one finds (see Appendix D),

$$\alpha_2 = 2 - \frac{\epsilon}{2}, \qquad \beta_2 = 2 - \frac{\epsilon}{3}. \tag{51}$$

Note that while here it happens that $\alpha_2 = 2\alpha$ and $\beta_2 = 2\beta$ at leading order in $\epsilon$, this is not generically true and is not expected to hold at higher orders in $\epsilon$. Comparing to Eq. (29), this result demonstrates a marked difference between the Schrödinger cat population dynamics and that of classical directed percolation.

To conclude this section, we observe a peculiarity of the RG flow equations Eq. (48). One may note that the parameter subspace in which all couplings are real is invariant. This occurs when there is no Hamiltonian (see Appendix A), which implies that Hamiltonian terms are never generated under RG if they are not already present microscopically. This turns out to have significant consequences for the fate of nonlinear couplings $\beta_1, \beta_2$ at large scales. Examining the microscopic form of the couplings (discussed in Appendix A), the bare values of $\beta_1 \beta_2$ is real, $Z(l=0) = 1$, and $D(l=0)$ may take different complex values but $\mathrm{Re}(D) > 0$ always. Thus, the imaginary component of $U(l=0)$ comes entirely from $D(l=0)$; if $D(l=0)$ is real then $U(l=0)$ is real. Examining the solutions to Eq. (49),

$$U(l) = \frac{U(0)e^{l\epsilon}}{1 + \frac{3C_d}{\epsilon}(1 - e^{\epsilon l})U(0)}, \tag{52}$$

one sees that if $U(0)$ has a finite imaginary part then $U(l \to \infty) = U^\star$, but if $U(0)$ is real then $U$ diverges to positive infinity as $l \to \infty$. Thus, if $D(l=0)$ is strictly real there is a pole in the RG flow equation and the $U^\star$ fixed point is inaccessible and the theory instead flows to strong coupling.

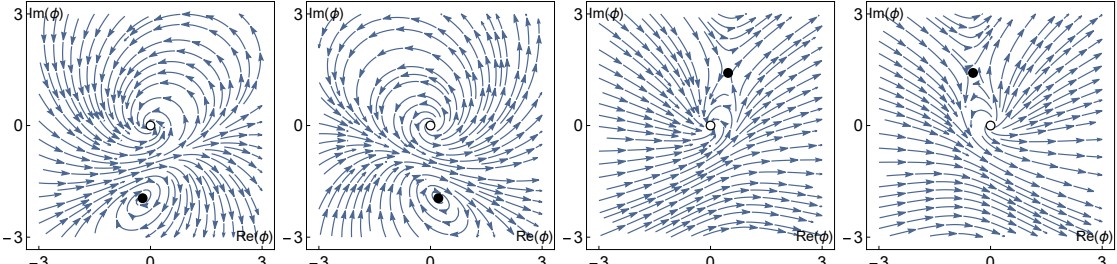

Figure 7: Plots of various trajectories of the mean-field dynamics of Eq. (42) demonstrating the stability of the fixed points for different dynamical regimes. In all plots, the horizontal and vertical axes correspond to the real and imaginary parts of $\phi$ and the white and black points show the dark and active fixed points. All plots are made with $|\alpha| = 2$ and $\beta_1 = 1$. The left two plots show the double sink and double source regimes as defined by Eq. (53). They have $\arg(\alpha) = 5\pi/8$ and $3\pi/8$ and $\beta_3 = \mp i/2$ respectively. The right two plots show the regime with $|\beta_3| > |\beta_1|$ in which the active fixed point is a saddle. They have $\arg(\alpha) = 3\pi/4$ and $\pi/4$ respectively and $\beta_3 = 2$.

## 4.3 Critical scaling for general couplings

We now consider the fully generic model with finite $\beta_3$. Microscopically, this additional term arises from cat-like superpositions of the competition and death processes, e.g. $\hat{L} \sim \hat{a}^2 + \hat{a}$, and cubic Hamiltonian terms. For $\beta_3 \neq 0$ the Reggeon symmetry, Eq. (46) is broken, but it is nevertheless possible to have a continuous transition with a critical behavior distinct from that of the complex Reggeon theory. Before discussing this, we first note that the existence of a continuous transition depends on the value of $\beta_3$, and for certain values of $\beta_3$ the transition can instead become the first order.

The mean-field dynamics, described by the spatially homogeneous version of Eq. (42), has three different dynamical regimes. The first two occur for $|\beta_3| < |\beta_1|$. For most values of $\beta_3$, the mean-field dynamics have the same behavior as the complex Regge theory: the origin is a source/sink and the active fixed point is a sink/source for $\text{Re}(\alpha) \gtrless 0$. For large values of $\text{Im}(\alpha)$ there is a bifurcation for certain values of $\beta_{1,3}$, the exact conditions for which are:

$$|\beta_1| \cos\left(\arg(\alpha)\right) \gtrless |\beta_3| \cos\left(\arg(\alpha) - \arg(\beta_1) - \arg(\beta_3)\right), \quad \text{Re}(\alpha) \lessgtr 0. \tag{53}$$

When this condition is satisfied, the stability of the active fixed point reverses and the dynamics has either two sources or two sinks depending on the sign of $\alpha$. Bringing $|\beta_3|$ equal to $|\beta_1|$ sends the active fixed point to infinity. When $|\beta_3| > |\beta_1|$ the active fixed point becomes again finite, but it is always a saddle. The two new dynamical regimes are depicted in Fig. 7; the right two panels of Fig. 5 shows mean field phase diagrams for $\beta_3$ finite for both $|\beta_1| \lessgtr 0|\beta_3|$.

Only the first of these these three situations results in a continuous transition. In either of the other two regimes, the active fixed point is dynamically unstable in the active phase, meaning the population diverges to infinity (or to some large but finite value if higher-order nonlinear terms are included to stabilize the theory; in this situation quartic terms can be understood as dangerously irrelevant). Thus, in crossing from the dark to active phase, the stationary population changes discontinuously and the transition is first order.

When a continuous transition is possible, there are no modifications to the flow equations of any of the couplings shown in Eq. (48) at leading order. The only addition is the RG flow equation for $\beta_3$, for which there are three diagrams that contribute (shown in Fig. 8. The flow equation is:

$$\partial_l \beta_3 = \left( d + z + \Delta_\phi + \bar{\Delta}_\phi + \bar{\Delta}_\chi + \frac{C_d \beta_1 \beta_2}{ZD^2} + \frac{C_d}{\bar{Z}D + Z\bar{D}} \left( \frac{\bar{\beta}_2 \beta_3}{\bar{D}} + \frac{\bar{\beta}_3 \beta_2}{D} \right) \right) \beta_3. \tag{54}$$

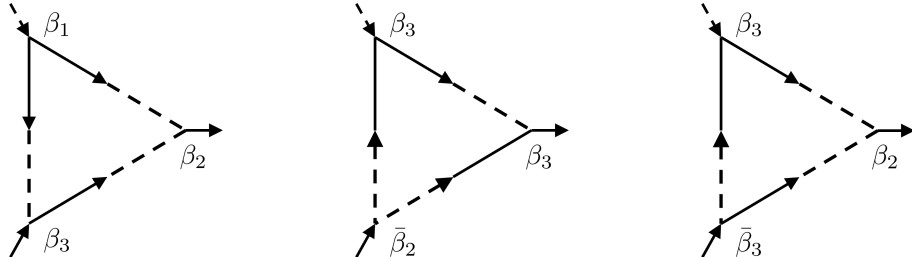

Figure 8: 1-loop diagrams contributing to the renormalization of $\beta_3$.

The critical surface of the complex Reggeon theory is identified as above using Eq. (49) $U^\star = -\epsilon/3C_d$ and $\alpha^\star = \beta_3^\star = 0$. However, linearizing Eq. (54) around any fixed point on this surface gives $\partial_l \beta_3 \simeq \epsilon \beta_3/3$. Thus the entire complex Reggeon critical surface is unstable to a $\beta_3$ perturbation for $\epsilon > 0$ and so a nonzero bare $\beta_3$ carries one away from the complex Reggeon theory towards a new fixed point with a finite $\beta_3^\star$.

A new set of fixed points with finite $\beta_3^\star$ is conveniently identified by writing the flow equations for the parameter $W = \bar{\beta}_2 \beta_3 / \bar{D}(\bar{Z}D + Z\bar{D})$,

$$\partial_l W = \big(\epsilon + C_d(2U + W + \bar{W})\big)W \, . \tag{55}$$

Together with Eq. (49) and their complex conjugates, these constitute a closed set of four RG flow equations. A new fixed point is found by putting $U^\star = -\epsilon/3C_d = W^\star + \bar{W}^\star$. Only three values of the 10 couplings $Z, D, \beta_{1,2,3}$ and their complex conjugates are fixed by this condition. There is critical surface of real dimension seven. A linear stability analysis shows that the critical surface is stable in all three directions point away in the space of the $Z, D, \beta_{1,2,3}$ and their complex conjugates (see Appendix C). Note however that similar to the complex Reggeon theory, for some regions of bare values of $U$ and $W$ (the conditions for which are discussed in Appendix A) the critical surface is inaccessible and the theory flows to a strong coupling.

The field scaling dimensions, $\Delta_{\phi,\chi}$, and the dynamical critical exponent $z$ may be obtained in the same way as discussed for the complex Reggeon theory and their values are the same as there. Because there is no modification to the RG equation for $\alpha$ from the inclusion of a finite $\beta_3$, the standard critical exponents are also unmodified. Equation (50) still holds, despite being at a different critical point. There are however modifications to the exponents governing the scaling of $n$ compared to Eq. (51). For finite $\beta_3$, one instead finds:

$$\alpha_2 = 2 - \frac{\epsilon}{6}, \qquad \beta_2 = 2 - \frac{\epsilon}{6} \, . \tag{56}$$

A derivation is presented in Appendix D. Note that, unlike the complex Reggeon theory, $\alpha_2 \neq 2\alpha$ and $\beta_2 \neq 2\beta$ already in linear order in $\epsilon$.

# 5 Discussion and conclusion

In this work, we established an exact correspondence between classical population dynamics and Lindbladians with the weak local $U(1)$ symmetry and the dark extinct state. This correspondence lead naturally to quantum generalizations of population dynamics in which this symmetry is not respected and the state of a system may consist of coherent superpositions of different population numbers. We also developed a field theory approach to studying quantum population models, which has a general utility for field theories with dark states. This method was applied to study a generic quantum population model of a single species, which

possesses a mix of classical birth and death processes and quantum Schrödinger cat population processes, which take the population number into a coherent superposition of different numbers. We found that this theory possesses a non-equilibrium dark state phase transition between a dark extinct phase and an active phase with a stable quantum population. From the perspective, where quantumness originates from the local symmetry breaking (and thus is a relevant perturbation), it is natural to expect that quantum population models belong to universality classes different from those of their classical counterparts. This is indeed what we found here for the model with no other constraints beyond supporting the dark state.

We note that this result differs from the several existing results on dark state phase transitions in models featuring mixtures of classical and quantum population processes [20,21,25]. In these works, a phase transition was identified as belonging to the classical tri-critical directed percolation universality class. This is however not in contradiction with our result. The bosonic theory studied in [25] has a global weak $U(1)$ symmetry and so falls into a different universality class than our Schrödinger cat population dynamics, which lacks such a symmetry.

Going forward, it may be interesting to more thoroughly study the dynamics of the Schrödinger cat field theory. Classical population dynamics typically demonstrate ballistic spreading of populations despite being diffusive at the linear level. This is due to nonlinear front solutions of the FKPP-like equations [72,73] describing the semiclassical dynamics of classical populations. The semiclassical dynamics of the Schrödinger cat theory Eqs. (47) and (42) are essentially complexified FKPP equations, and so likely support qualitatively similar nonlinear wave solutions. The existence of such solutions would have the largest effect on dynamics in low spatial dimensions, and could lead to modifications of, for example, extinction times.

The approach developed here could be used to various other quantum population models. In classical population and reaction models, the addition of symmetries or additional dynamical constraints lead to a variety of different universality classes [68,79–81], which may be classified according to the topology of their semiclassical phase portrait [50]. A similar classification scheme may be possible in the quantum setting, though it would require the inclusion of additional classes based on dynamic constraints that are distinctly quantum, as exemplified at the end of Section 2.

It may also be interesting to examine quantum population models with multiple species, such as quantum versions of predator-prey or rock-paper-scissors dynamics. Generic absorbing state transitions in classical population models with multiple species are thought to belong to the same directed percolation universality class as the single species Reggeon theory, but may they may develop critical behavior different from directed percolation at multi-critical points [82–84]. It seems plausible that a similar situations may occurs for multi-species quantum population and quantum reaction models and is worth further exploration in future work. In a more general context, the Reggeon field theory and the above Schrödinger cat field theory represent generic theories obeying absorbing state constraints (Eq.s (26) and (27) and Eq.s (39) and (40) resp.) with a single (real resp. complex) scalar degree of freedom. More general models with vector or matrix valued fields obeying similar absorbing state conditions have, to our knowledge, not been thoroughly studied and may present interesting new classes of field theories with potentially new critical behavior and dynamics. In particular, it may be interesting to see if such models admit simplified descriptions in the limit of large numbers of degrees of freedom, as is known to occur in certain families of classical reaction-diffusion models [79].

As noted in the main text, there is a formal similarity between the dark state condition, Eq. (35), and the equilibrium FDT, which similarly imposes a fixed relation between Keldysh and spectral components of the Green's function. The equilibrium FDT condition, and the other relations between $n$-point functions that must hold in equilibrium as consequence of

KMS, can be cast in the form of a symmetry of the Keldysh action [85]. It is possible that a similar symmetry description underpins relations between $n$-point functions in theories with the dark states. If such a description does exist, this could lead to a new way to use symmetry to classify non-equilibrium theories with absorbing/dark states. Somewhat similar ideas were recently put forward for both classical [86] and quantum [87, 88] non-equilibrium theories with generic stationary states which involve generalized concepts of detailed balance. Dark state models represent the opposite extreme, in which detailed balance is maximally violated.

Finally, it would be interesting to further explore what utility quantum reaction models may have as protocols for preparing and protecting entangled quantum states. Keeping in mind existing proposals to realize certain quantum states with dissipation make use of either dynamics with dark spaces [34–36] or classical reaction models [38,39], quantum population dynamics language would seem to be a natural language with which look for for generalizations of or improvements to existing proposals. More generally, the relationship between dark state field theories (and the phase transitions they may posses) and the entanglement properties of the dark space would be an interesting direction for future study.

## Acknowledgments

The authors are grateful for useful discussions with Sebastian Diehl, Mark Dykman, and Yuval Gefen.

**Funding information** This work was supported by the NSF grant DMR-2338819.

## A Microscopic parameters

In this appendix, we consider the microscopic origin of the terms in the Schrödinger cat population dynamics. The general form of jump operators representing quantum population processes is $\hat{L}_v = \hat{O}_v \hat{a}$ where $\hat{O}_v$ can be any operator. Up to second order in Bose operators, generic jump operators have the form:

$$\hat{L}_v = (l_{1v}\hat{a} + l_{2v}\hat{a}^\dagger + l_{3v})\hat{a} + \dots \tag{A.1}$$

Each such jump operator corresponds to a superposition of up to three reaction rules: the two classical rules $A \to \emptyset$ and $2A \to \emptyset$, and the quantum process $\hat{L} = \hat{a}^\dagger \hat{a}$ that does not encode any classical reaction process. One can also add a Hamiltonian, which has the general form up to the fourth order in Bose operators:

$$\hat{H} = h_2\hat{a}^\dagger\hat{a} + \hat{a}^\dagger(\bar{g}_3\hat{a}^\dagger + g_3\hat{a})\hat{a} + \hat{a}^\dagger(h_4\hat{a}^\dagger\hat{a} + \bar{g}_4\hat{a}^{\dagger 2} + g_4\hat{a}^2)\hat{a} + \dots \tag{A.2}$$

Translating this to an action Eq. (38) results in a theory with the effective Hamiltonian of the form:

$$K = \bar{\chi}\phi(\alpha + \beta_1\phi + \beta_2\bar{\chi} + \beta_3\bar{\phi} + \eta_1\phi^2 + \eta_2\phi\bar{\chi} + \eta_3\bar{\chi}^2 + \eta_4\phi\bar{\phi} + \eta_5\bar{\chi}\bar{\phi} + \eta_6\bar{\phi}^2 + \zeta\bar{\phi}\chi) + \text{c.c.}, \tag{A.3}$$

where the coefficients have the following relations to microscopic parameters:

$$\alpha = -ih_2 - \frac{1}{2}\sum_\nu \bar{l}_{3\nu} l_{3\nu}, \tag{A.4a}$$

$$\beta_1 = -\frac{i}{\sqrt{2}} g_3 - \frac{1}{2\sqrt{2}}\sum_\nu (\bar{l}_{3\nu} l_{1\nu} + \bar{l}_{2\nu} l_{3\nu}), \tag{A.4b}$$

$$\beta_2 = -i\sqrt{2}\bar{g}_3 - \frac{1}{\sqrt{2}}\sum_\nu (\bar{l}_{3\nu} l_{2\nu} + \bar{l}_{1\nu} l_{3\nu}), \tag{A.4c}$$

$$\beta_3 = -i\sqrt{2}\bar{g}_3 - \frac{1}{\sqrt{2}}\sum_\nu \bar{l}_{1\nu} l_{3\nu}, \tag{A.4d}$$

$$\eta_1 = -\frac{i}{2} g_4 - \frac{1}{4}\sum_\nu \bar{l}_{2\nu} l_{1\nu}, \tag{A.4e}$$

$$\eta_2 = -ih_4 - \frac{1}{2}\sum_\nu (\bar{l}_{1\nu} l_{1\nu} + \bar{l}_{2\nu} l_{2\nu}), \tag{A.4f}$$

$$\eta_3 = -2i\bar{g}_4 - \sum_\nu \bar{l}_{1\nu} l_{2\nu}, \tag{A.4g}$$

$$\eta_4 = -ih_4 - \frac{1}{2}\sum_\nu \bar{l}_{1\nu} l_{1\nu}, \tag{A.4h}$$

$$\eta_5 = -3i\bar{g}_4 - \frac{3}{2}\sum_\nu \bar{l}_{1\nu} l_{2\nu}, \tag{A.4i}$$

$$\eta_6 = -\frac{3i}{2}\bar{g}_4 - \frac{1}{4}\sum_\nu \bar{l}_{1\nu} l_{2\nu}, \tag{A.4j}$$

$$\zeta = \frac{1}{2}\sum_\nu \bar{l}_{2\nu} l_{2\nu}, \tag{A.4k}$$

where the rate $\gamma_\nu$ of each jump operator $\hat{L}_\nu$ has for convenience been absorbed into the coefficients $l_{j\nu}$.

When extended to finite dimensions, there are two different types of quantum hopping terms at lowest order: coherent and incoherent. One involves an addition to the Hamiltonian and the other involves a new set of jump operators. These are most easily implemented on lattice using nearest-neighbor hopping,

$$\hat{H}_D = 2u \sum_{\mathbf{r},i} \left( \hat{a}^\dagger_{\mathbf{r}+\mathbf{a}_i} \hat{a}_{\mathbf{r}} + \hat{a}^\dagger_{\mathbf{r}-\mathbf{a}_i} \hat{a}_{\mathbf{r}} + \text{h.c.} \right), \tag{A.5a}$$

$$\hat{L}_{D,\mathbf{r},i,\pm} = \sqrt{2v} \left( \hat{a}_{\mathbf{r}\pm\mathbf{a}_i} - \hat{a}_{\mathbf{r}} \right), \tag{A.5b}$$

where $\mathbf{a}_i$ are the lattice unit vectors. In the long-wavelength limit on a square lattice, these terms lead to the complex diffusion term in the action, $\bar{\chi} D \partial^2_{\mathbf{r}} \phi + \text{c.c.}$, with the complex diffusion coefficient given by $D = u + iv$.

## B  Derivation of RG flow equations

This section discusses details of the perturbative RG calculations performed to arrive at Eq. (48). This is done using a standard $\epsilon$-expansion approach. The details are very similar to those of the classical directed percolation RG, see, e.g., Ref. [52].

To begin, we impose a momentum cutoff $|\mathbf{k}| < \Lambda$, and subsequently rescale the spacetime coordinates and fields in the standard way:

$$\mathbf{r} \to b\mathbf{r}, \qquad t \to b^z t, \qquad \phi \to b^{\Delta_\phi} \phi, \qquad \chi \to b^{\Delta_\chi} \chi. \tag{B.1}$$

The quantities $\Delta_\phi$ and $\Delta_\chi$ are the scaling dimensions of the classical and auxiliary fields, $\phi$ and $\chi$, and $z$ is the dynamical critical exponent. One may then split each field into a sum of slow and fast modes $\phi(\epsilon, \mathbf{k}) = \phi_s(\epsilon, \mathbf{k}) + \phi_f(\epsilon, \mathbf{k})$ where $\phi_s$ is supported for $|\mathbf{k}| < \Lambda$ and $\phi_f$ is supported for $\Lambda < |\mathbf{k}| < b\Lambda$. With this, the action can be expressed as $S = S_s + S_f + S_{s-f}$ where $S_s, S_f$ consist of only the slow/fast fields resp. and $S_{s-f}$ contains the nonlinear cross terms with both types of fields. To facilitate calculations, we introduce the coupling $Z$ as the name of the coupling of the kinetic term $\bar{\chi}\partial_t \phi$, which always possesses the bare value $Z = 1$. With this, the bare Green's function acquires the form:

$$\mathcal{G}(p) = \frac{1}{-Zi\epsilon + D\mathbf{k}^2 - \alpha}, \tag{B.2}$$

where $p = (\epsilon, \mathbf{k})$ is the $d + 1$-momentum.

Next, the fast fields are averaged over. The leading-order corrections to the propagator are given at the second order,

$$\frac{1}{2}\langle S_{s-f}^2 \rangle_f = \left\langle \int \beta_1 \bar{\chi}_s \phi_f \phi_f \int \beta_2 \phi_s \bar{\chi}_f \bar{\chi}_f \right\rangle_f. \tag{B.3}$$

Performing the Gaussian averaging over the fast modes, one obtains the following modification to the action upon re-exponentiation,

$$S_s \to S_s - 2\beta_1\beta_2 \int \mathrm{d}p\, \bar{\chi}_s(p)\phi_s(p) \int \mathrm{d}p_f\, \mathcal{G}(p_f^+)\mathcal{G}(-p_f^-), \tag{B.4}$$

where $\mathcal{G}$ is given by Eq. B.2 and $p_f^\pm = p_f \pm p/2$. This correction corresponds to the diagram for the propagator shown in the main text in Fig. 6. After integrating over $\epsilon_f$ and expanding to leading order in all couplings, the correction to the action is:

$$-\frac{\beta_1\beta_2}{Z} \int \frac{\mathrm{d}\mathbf{k}_f}{(2\pi)^d} \left( \frac{1}{D\mathbf{k}_f^2} - \left( \frac{D}{4}\mathbf{k}^2 - \frac{iZ}{2}\epsilon - \alpha \right) \frac{1}{(D\mathbf{k}_f^2)^2} + \dots \right). \tag{B.5}$$

The first term renormalizes $\alpha$ and will subsequently be absorbed. The second term results in corrections to $Z, D$, and $\alpha$. Performing the integral over $\mathbf{k}_f$, the entirety of the RG procedure results in the replacements:

$$Z \to \left( b^{d+\Delta_\phi+\bar{\Delta}_\chi} + \frac{C_d\beta_1\beta_2}{2ZD^2(d-4)}(b^{d-4}-1) \right)Z, \tag{B.6a}$$

$$D \to \left( b^{d+z-2+\Delta_\phi+\bar{\Delta}_\chi} + \frac{C_d\beta_1\beta_2}{4ZD^2(d-4)}(b^{d-4}-1) \right)D, \tag{B.6b}$$

$$\alpha \to \left( b^{d+z+\Delta_\phi+\bar{\Delta}_\chi} + \frac{C_d\beta_1\beta_2}{ZD^2(d-4)}(b^{d-4}-1) \right)\alpha, \tag{B.6c}$$

with $C_d = \Lambda^{d-4}2^{1-d}\pi^{-d/2}/\Gamma(d/2)$. Performing an infinitesimal scaling transformation by the replacement $b = 1 + l$ and differentiating, one obtains the first three of Eqs. (48).

For corrections to the vertices, we focus on the corrections to $\beta_1$; the corrections to $\beta_2$ are given by swapping $\beta_1 \leftrightarrow \beta_2$ in all calculations. The leading order correction to $\beta_1$ comes at third order,

$$-\frac{1}{3!}\langle S_{s-f}^3 \rangle_f = 2\left\langle \int \beta_1 \bar{\chi}_s \phi_f \phi_f \int \beta_1 \phi_s \bar{\chi}_f \phi_f \int \beta_2 \phi_s \bar{\chi}_f \bar{\chi}_f \right\rangle_f. \tag{B.7}$$

Averaging the fast modes gives a correction to the action:

$$S_s \to S_s - 8\beta_1^2\beta_2 \int \mathrm{d}p_1 \mathrm{d}p_2 \mathrm{d}p_3\, \bar{\chi}_s(p_1)\phi_s(p_2)\phi_s(p_3)\delta(p_1-p_2-p_3) \int \mathrm{d}p_f\, \mathcal{G}(p_f)\mathcal{G}(p_1-p_f)\mathcal{G}(p_f-p_2), \tag{B.8}$$

where only the terms contributing to the renormalization of $\beta_1$ are written. The corresponding diagram is shown in Fig. 6 in the main text. All other 1-loop diagrams contributing the $\beta_1$ vanish by causality. Next, one may expand the resulting vertex function in the slow momenta, keeping only the part with zero momentum transfer $\mathcal{G}(p_f)\mathcal{G}(p_1-p_f)\mathcal{G}(p_f-p_2) \simeq \mathcal{G}(-p_f)\mathcal{G}(p_f)^2$. Subsequent integrating over $p_f$ then gives the 1-loop correction to the vertices,

$$\beta_1 \to \left( b^{d+z+2\Delta_\phi+\bar{\Delta}_\chi} + \frac{2C_d\beta_1\beta_2}{ZD^2(d-4)}(b^{d-4}-1) \right)\beta_1, \tag{B.9a}$$

$$\beta_2 \to \left( b^{d+z+\Delta_\phi+2\bar{\Delta}_\chi} + \frac{2C_d\beta_1\beta_2}{ZD^2(d-4)}(b^{d-4}-1) \right)\beta_2, \tag{B.9b}$$

which upon differentiation after putting $b = 1 + l$ reproduces the remaining two equations from Eq. (48). The form of these equations clearly indicates a critical dimensions of $d_c = 4$.

The inclusion of the $\beta_3$ in the fully general theory does not produce new 1-loop diagrams for any of the other couplings and thus dues not lead to any additional corrections at leading order. There are three different terms that contribute to the renormalization of $\beta_3$ itself, which originate from the third order terms in the action,

$$
\begin{aligned}
-\frac{1}{3!}\langle S_{s-f}^3\rangle_f = &-\left\langle \int \beta_1\bar{\chi}_s\phi_f\phi_f \int \beta_2\phi_s\bar{\chi}_f\bar{\chi}_f \int \beta_3\bar{\phi}_s\bar{\chi}_f\phi_f \right\rangle_f \\
&-\left\langle \int \beta_3\phi_s\bar{\chi}_f\bar{\phi}_f \int \beta_3\bar{\chi}_s\phi_f\bar{\phi}_f \int \bar{\beta}_2\bar{\phi}_s\chi_f\chi_f \right\rangle_f + \text{c.c.},
\end{aligned}
\tag{B.10}
$$

where only the terms contributing to the renormalization of $\beta_3$ are written.

After averaging over the fast modes the first term leads to a modification to the action:

$$4\beta_1\beta_2\beta_3\int dp_1 dp_2 dp_3\, \bar{\chi}_s(p_1)\phi_s(p_2)\bar{\phi}_s(p_3)\delta(p_1-p_2-p_3)\int dp_f \mathcal{G}(p_f)\mathcal{G}(p_1-p_f)\mathcal{G}(p_2-p_f). \tag{B.11}$$

Keeping only the part with zero momentum transfer, this leads to the same integral expression as appears in the renormalization for $\beta_{1,2}$ up to a factor of $1/2$. The second term becomes after averaging:

$$2\bar{\beta}_2\beta_3^2\int dp_1 dp_2 dp_3\, \bar{\chi}_s(p_1)\phi_s(p_2)\bar{\phi}_s(p_3)\delta(p_1-p_2+p_3)\int dp_f \mathcal{G}(p_f)\bar{\mathcal{G}}(p_1+p_f)\bar{\mathcal{G}}(p_2-p_f). \tag{B.12}$$

The zero momentum part is $\mathcal{G}(p_f)\bar{\mathcal{G}}(p_1 + p_f)\bar{\mathcal{G}}(p_2 - p_f) \simeq \mathcal{G}(p_f)\bar{\mathcal{G}}(p_f)\bar{\mathcal{G}}(-p_f)$; integrating this term over $p_f$ gives a factor of $C_d/2\bar{D}(\bar{D}Z + D\bar{Z})$. Taking all three terms together, this gives the the 1-loop correction to $\beta_3$,

$$\beta_3 \to \left( b^{d+z+\Delta_\phi+\bar{\Delta}_\chi+\Delta_{\bar{\phi}}} + \frac{C_d(b^{d-4}-1)}{Z(d-4)}\left( \frac{\beta_1\beta_2}{D^2} + \frac{\bar{\beta}_2\beta_3}{\bar{D}(\bar{D}Z + D\bar{Z})} + \frac{\beta_2\bar{\beta}_3}{D(\bar{D}Z + D\bar{Z})} \right) \right)\beta_3, \tag{B.13}$$

which upon differentiation after putting $b = 1 + l$ reproduces Eq. (54).

# C Stability of the critical surface

Linearizing the RG flow equations (48) around any given fixed point on the critical surface determines its stability. To this end, for each coupling $g$ we put $g(l) = g^\star + \delta_g(l)$ and expand to leading order in each $\delta_g$. This may be performed for a general point on the critical surface

$\alpha^\star = 0$ and $(Z^\star, D^\star, \beta_1^\star, \beta_2^\star)$ with $U^\star = \beta_1^\star \beta_2^\star / Z^\star D^{\star 2} = -\epsilon/3C_d$ and $\beta_2/\beta_1$ fixed by the Regge symmetry. The linearized RG equations are then:

$$\partial_l \alpha \simeq \left(2 - \frac{\epsilon}{4}\right)\alpha, \tag{C.1a}$$

$$\partial_l \delta_Z \simeq \frac{\epsilon}{6}\delta_Z + \frac{\epsilon Z^\star}{3D^\star}\delta_D - \frac{\epsilon Z^\star}{6\beta_1^\star}\delta_{\beta_1} - \frac{\epsilon Z^\star}{6\beta_2^\star}\delta_{\beta_2}, \tag{C.1b}$$

$$\partial_l \delta_D \simeq \frac{\epsilon D^\star}{12Z^\star}\delta_Z + \frac{\epsilon}{6}\delta_D - \frac{\epsilon D^\star}{12\beta_1^\star}\delta_{\beta_1} - \frac{\epsilon D^\star}{12\beta_2^\star}\delta_{\beta_2}, \tag{C.1c}$$

$$\partial_l \delta_{\beta_1} \simeq \frac{2\epsilon \beta_1^\star}{3Z^\star}\delta_Z + \frac{4\epsilon \beta_1^\star}{3D^\star}\delta_D - \frac{2\epsilon}{3}\delta_{\beta_1} - \frac{2\epsilon \beta_1^\star}{3\beta_2^\star}\delta_{\beta_2}, \tag{C.1d}$$

$$\partial_l \delta_{\beta_2} \simeq \frac{2\epsilon \beta_2^\star}{3Z^\star}\delta_Z + \frac{4\epsilon \beta_2^\star}{3D^\star}\delta_D - \frac{2\epsilon \beta_2^\star}{3\beta_1^\star}\delta_{\beta_1} - \frac{2\epsilon}{3}\delta_{\beta_2}, \tag{C.1e}$$

and their complex conjugates. The critical surface is unstable in the direction of the complex mass $\alpha$, which is independent of the other couplings at the linear level. Upon diagonalizing the coupling matrix of the latter four equations, one finds a single non-zero irrelevant eigenvalue $-\epsilon$ regardless of the values of the $(Z^\star, D^\star, \beta_1^\star, \beta_2^\star)$. The other three eigenvalues are zero (i.e. marginal) and their eigenvectors correspond to motion parallel to the critical surface, or to different values of $\beta_2/\beta_1$ ratio.

For finite $\beta_3$, one must also include Eq. (54). Its linearization around the fixed point with non-zero $\beta_3^\star$ yields:

$$\partial_l \delta_{\beta_3} \simeq \left(\frac{\epsilon}{3Z^\star} + \frac{\epsilon \bar{D}^{\star 2} W^\star}{3\bar{\beta}_2^\star \beta_3^\star}\right)\delta_Z + \left(\frac{2\epsilon - 3C_d \bar{W}^\star}{3\bar{D}^\star} + \frac{\epsilon \bar{Z}^\star \bar{D}^\star W^\star}{\bar{\beta}_2^\star \beta_3^\star}\right)\delta_D - \frac{\epsilon}{3\beta_1^\star}\delta_{\beta_1} - \frac{\epsilon - 3C_d \bar{W}^\star}{3\beta_2^\star}\delta_{\beta_2}$$
$$- \frac{\epsilon - 3C_d W^\star}{3\beta_3^\star}\delta_{\beta_3} + \frac{\epsilon D^{\star 2}\bar{W}^\star}{3\beta_2^\star \bar{\beta}_3^\star}\bar{\delta}_Z + \left(\frac{\epsilon Z^\star D^\star \bar{W}^\star}{3\beta_2^\star \bar{\beta}_3^\star} - \frac{C_d W^\star}{\bar{D}^\star}\right)\bar{\delta}_D + \frac{C_d W^\star}{\bar{\beta}_2^\star}\bar{\delta}_{\beta_2} + \frac{C_d \bar{W}^\star}{\bar{\beta}_3^\star}\bar{\delta}_{\beta_3}. \tag{C.2}$$

Diagonalizing the 10-dimensional coupling matrix resulting from this equation and its complex conjugate together with Eq. (C.1) gives three nonzero eigenvalues. The values of these eigenvalues is the same for all values on the critical surface. Two of these have the value $-\epsilon$ and correspond to the perturbations away from the critical surface in the direction of $U$ and $\bar{U}$. The third has the value $-\epsilon/3$ and it involves perturbations away from the critical surface in a direction with finite $\beta_3$. The other seven eigenvalues are all zero and correspond to perturbations in directions parallel to critical surface. This demonstrates stability of the critical manifold to linear order in $\epsilon$.

# D    Critical exponents

The values of the dynamical critical exponent $z$ and the field scaling dimensions $\Delta_\phi$ and $\Delta_\chi$ are obtained in the main text. The inverse of the correlation length exponent $\nu$ is equal to the critical scaling of the linear death rate, $[\alpha] = 1/\nu$. Here $[\alpha]$ denotes the scaling of $\alpha$ under the RG transformation, $\alpha \to b^{[\alpha]}\alpha$. This may be obtained from the linearized flow of $\alpha$ near the critical point, $\partial_l \alpha \simeq \alpha/\nu$. Examining Eq. (C.1) one finds $1/\nu = 2 - \epsilon/4$, from which follows the value of $\nu$ in Eq. (50). The remaining two exponents $\alpha$ and $\beta$ are obtained from scaling relations using $\nu$ and $\Delta_\phi$. The first is defined as the long-time scaling of the order parameter $\phi$ at the critical point as a function of time, $\langle \phi(t) \rangle \sim t^{-\alpha}$; the other is defined as the scaling of the stationary value of $\phi$ near the critical point as a function of the distance from the critical point, measured using the complex mass, $\langle \phi \rangle \sim |\alpha|^\beta$. Performing the RG rescaling from Eq. (B.1) on

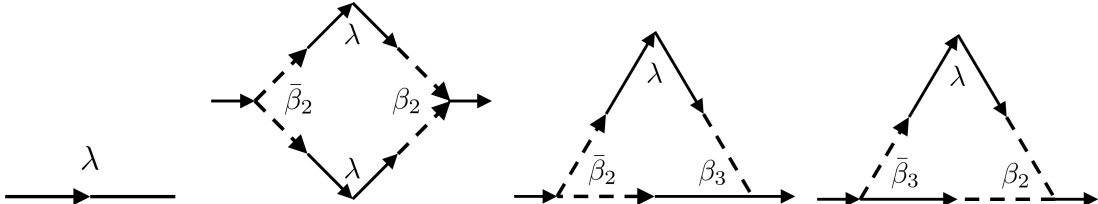

Figure 9: Diagrammatic depiction of auxiliary vertex and the three diagrams contributing to the renormalization of the auxiliary coupling $\lambda$.

either side of both expressions and keeping in mind that the scaling of $\alpha$ is set by $\nu$ near the critical point, one finds the relations:

$$\alpha = -\Delta_\phi/z\,, \qquad \beta = -\nu\Delta_\phi\,. \tag{D.1}$$

Using the values obtained in the main text $\Delta_\phi = -2 + 7\epsilon/12 = \Delta_\chi$ and $z = 2 - \epsilon/12$, the above relations reproduce the remaining two exponents in Eq. (50).

To determine the critical points associated with the population $n \sim \bar\phi\phi$ from Eq. (51) we use the fact that under the RG transformation one has $n \to b^{\Delta_{\bar\phi\phi}} n$, where $\Delta_{\bar\phi\phi}$ is the scaling dimension of the composite operator $n$. Defining the critical scaling of $n$ analogous to those of $\phi$ discussed above, $\langle n(t)\rangle \sim t^{-\alpha_2}$ and $\langle n \rangle \sim |\alpha|^{\beta_2}$, one has by a similar argument the scaling relations:

$$\alpha_2 = -\Delta_{\bar\phi\phi}/z\,, \qquad \beta_2 = -\nu\Delta_{\bar\phi\phi}\,. \tag{D.2}$$

Thus the task of determining $\alpha_2$ and $\beta_2$ reduces to determining the scaling dimension of $\bar\phi\phi$.

Because the fixed point is non-Gaussian, one should generically expect $\Delta_{\bar\phi\phi} \neq \bar\Delta_\phi + \Delta_\phi$. To determine the scaling dimension, one may introduce an additional coupling associated with $\bar\phi\phi$ [78],

$$S \to S + \int \mathrm{d}x\,\lambda\bar\phi\phi\,. \tag{D.3}$$

The scaling of the new coupling $[\lambda]$ can be related to the scaling dimension of the corresponding operator by recalling that the action should remain invariant under RG,

$$\Delta_{\bar\phi\phi} = [\lambda] - d - z\,. \tag{D.4}$$

The inclusion of this new coupling modifies the RG flow equations, from which $[\lambda]$ can be deduced by linearizing around the modified critical point.

The inclusion of this new coupling can be thought of as an anomalous two-point vertex. This modifies the linear theory by breaking the dark state condition and giving a finite value to the anomalous Green's function $\langle\chi\bar\chi\rangle$, for which we will use the symbol $\mathcal{F}$. At the linear level, one has:

$$\mathcal{F}(p) = -\lambda\bar{\mathcal{G}}(p)\mathcal{G}(p)\,. \tag{D.5}$$

This addition causes a change to the diagrammatic rules and leads to a single new non-vanishing 1-loop diagram, as depicted in Fig. 9. This diagram renormalizes $\lambda$. Even before any calculation however, one may observe that this diagram contains two copies of the auxiliary vertex and thus two powers of $\lambda$. As a consequence, the only correction to $\lambda$ will occur at second order in $\lambda$ thus leading to the RG flow equation with the schematic form,

$$\partial_l\lambda = \left(d + z + \Delta_\phi + \bar\Delta_\phi\right)\lambda + O(\lambda^2)\,. \tag{D.6}$$

When linearizing around a point on the $\lambda^\star = 0$ critical surface, the second order terms will not contribute and thus there is no modification to the bare scaling. One thus finds $\partial_l \lambda \simeq (2 + \epsilon/12)\lambda$ and therefore $[\lambda] = 2 + \epsilon/12$, which along with Eq.s (D.2) and (D.4) reproduces Eq. (51).

The inclusion of finite $\beta_3$ leads to two more diagrams, both of which are only first order in $\lambda$, Fig. 9, and thus do contribute to the linearized flow near the critical surface. These contributions are obtained at second order in the action,

$$\frac{1}{2}\langle S^2_{\text{s-f}}\rangle_{\text{f}} = 2\left\langle \int \bar{\beta}_3 \bar{\phi}_{\text{s}} \phi_{\text{f}} \chi_{\text{f}} \int \beta_2 \phi_{\text{s}} \bar{\chi}_{\text{f}} \bar{\chi}_{\text{f}} \right\rangle_{\text{f}} + \text{c.c.}, \tag{D.7}$$

where only the terms contributing to the linearized flow of $\lambda$ are written. The first term leads to the modification to the action of the form

$$-4\bar{\beta}_3 \beta_2 \int \mathrm{d}p\, \bar{\phi}_{\text{s}}(p)\phi_{\text{s}}(p) \int \mathrm{d}p_{\text{f}}\, \mathcal{G}(p_{\text{f}}^+)\mathcal{F}(-p_{\text{f}}^-), \tag{D.8}$$

where $p_{\text{f}}^\pm = p_{\text{f}} \pm p/2$. Integrating over $p_{\text{f}}$ and keeping only the part with zero momentum transfer and performing the same calculation for the conjugated diagram, one finds the RG flow equation for the auxiliary coupling:

$$\partial_l \lambda = \left( d + z + \mathbf{\Delta}_\phi + \bar{\mathbf{\Delta}}_\phi - \frac{2C_d}{\bar{Z}D + Z\bar{D}}\left(\frac{\bar{\beta}_2\beta_3}{\bar{D}} + \frac{\bar{\beta}_3\beta_2}{D}\right)\right)\lambda. \tag{D.9}$$

Linearization around a point on the finite $\beta_3$ critical surface gives $\partial_l \lambda \simeq (2 - 7\epsilon/12)\lambda$ and therefore $[\lambda] = 2 - 7\epsilon/12$, which together with Eq.s (D.2) and (D.4) gives Eq. (56).

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
