# Peer review of "Population Dynamics of Schrödinger Cats"

_SciPost Physics, doi:SciPost Phys. 18, 046 (2025)_

## Round 2 · Referee Report · Anonymous (Referee 1) · 2024-10-10

Strengths

  1. interesting results relating quantum and classical population dynamics
  2. pedagogical presentation of the results

Weaknesses

no weakness

Report

The work “Population Dynamics of Schrodinger Cats” is devoted to study of a specific bosonic Keldysh field theory corresponding to extension of classical population dynamics to include non-classical processes. The paper is interesting and timely. The manuscript provide a novel and synergetic link between Linbladian dynamics of bosonic quantum systems (described in terms of Keldysh part integral) and classical population dynamics - an area well-studied previously. The paper is written in pedagogical style with many details simplifying the understanding of the matter. I strongly recommend publication of the manuscript in SciPost. Before publication I suggests for authors to consider the following comments:

i) The absence of some qubic terms, e.g. \bar\chi \chi \phi in Eq. (40) is discussed in the footnote on page 17. In particular, there is a claim that such terms are not generated by RG procedure. In my opinion, a bit more detailed discussion of this point would be beneficial for a reader. Is some symmetry that forbids emergence of such terms in the course of RG?

ii) The one-loop diagrams, responsible for renormalization of vertices \beta_1 and \beta_2 are shown In Fig. 5. It would be useful to connect these diagrams with diagrams (processes) in terms of reaction processes. It could make physics behind renormalization of the Keldysh action more transparent.

iii) As we know in \phi^4 field theory we can consider a complex field \phi to become a N-dimensional vector. It will affect the theory, in general, and RG equation in particular. Do such extensions are possible and meaningful for the Keldysh action (40)? If yes, is it possible to develop a kind of 1/N expansion? I understand that, perhaps, a detailed answer needs to do a separate work, but, in my opinion, a brief discussion of this issue would be useful for a reader.

Requested changes

some optional amendments are possible, see the report's items (i)-(iii)

Recommendation

Ask for minor revision

---

## Round 2 · Referee Report · Anonymous (Referee 2) · 2024-10-28

Strengths

1) interesting results in a timely problem 2) well written, self-contained discussion

Weaknesses

None

Report

In the manuscript, the authors establish an exact relation between classical population dynamics and Lindbladian evolution with a weak local U(1) symmetry and the dark extinct state. They use this relation to construct a quantum generalization of classical population dynamics by breaking this U(1) symmetry, thus allowing for superpositions of states with different occupation numbers. The authors then derive an effective field theory description (a "complex" Reggeon theory) and focus on the out-of-equilibrium phase transitions between an "absorbing" dark extinct state and an "active" phase with a finite occupation number. They illustrate their general considerations by analyzing a specific single-species case and calculating the critical exponents of the transition by means of the epsilon expansion.

The manuscript is well-organized, and the authors provide a comprehensive and self-contained introduction to the problem, clarifying all technical steps. The calculations appear valid, and the presentation style is quite clear. In my opinion, this work constitutes important progress in our understanding of quantum non-equilibrium dynamics with dark states and has potential for future extensions.

I have a few questions and suggestions for the authors to consider:

1) Is it expected that the mean-field theory becomes exact in the present context (both for the transition and the neighboring phases) above the upper critical dimension? In particular, does the active phase persist in d>4 for arbitrarily small positive Re(\alpha)>0, as the FKPP Eq.(46) seems to suggest? Some studies suggest that even for classical reaction-diffusion branching/annihilating dynamics (which share similar FKPP mean-field dynamics with Eq.(46) but with real parameters and \alpha>0), the absorbing phase transition occurs at a finite critical ratio of \alpha and \beta_1, across all dimensions. See, for instance, arXiv:0309504 and arXiv:0511456. Could the authors comment on whether such finite-coupling effects might also arise in their theory? 2) If I understood correctly, based on Eq.(57-59), any quartic interaction coupling in the Hamiltonian eventually corresponds to an RG irrelevant operator near the transition. Is there an intuitive physical argument for why interactions become irrelevant in high dimensions, and what role might they play in lower-dimensional systems? Could these interaction couplings be dangerously irrelevant? 3) As a minor point, I suggest explicitly labeling the axes (and the limits) in Fig. 4 and Fig. 6 to improve readability.

To summarize, I recommend the manuscript for publication in SciPost Physics after minor revision.

Requested changes

See points (1-3) in the report.

Recommendation

Ask for minor revision

---

## Round 2 · Referee Report · Anonymous (Referee 3) · 2024-11-4

Report

This authors creatively considers the population dynamics in the quantum realm where quantum coherences are also relevant. The rather striking conclusion of the paper is that the quantum analog of population dynamics with an absorbing state transition leads to new universal behavior distinct from the classical counterpart which is believed to describe the universality class of directed percolation.

The paper, including the conceptual ideas, the technical apparatus, and the narrative, is quite creative and elegant, and the results are rather striking. At the same time, there are some conceptual and quite a few technical points that need to be further clarified. I describe these points below.

1. Weak symmetry breaking: It is noted that breaking a weak symmetry is a relevant perturbation (in the sense of RG) and will give rise to new universal behavior. I am not sure if this conclusion is completely general. If we consider a classical rate equation for example, we can similarly write it as diagonal elements in a quantum representation. One can then consider quantum dynamics that doesn’t commute with the classical basis. Does this give rise to new universality classes? I think generically not. Indeed, this is often the fate of driven-dissipative systems where, despite the quantum dynamics, an effective thermal/classical behavior emerges. (I think the authors should also cite a few relevant papers on this point.) Another example would be perhaps a similar model as considered by the authors but assuming the transition from an absorbing state that is not completely fluctuationless (that is, consider a mixed state). I think in that case too quantum coherences and breaking the corresponding weak symmetry will not play an important role.
2. Field theory presentation: The authors do a good job for a light introduction into the field theory making the manuscript easy and enjoyable to read. This is at the expense of multiple references to a recent review paper by the same authors. Therefore, certain technical steps are not explained for a technically curious reader. Here are a few examples: in the field theory developed around Eq. 30, it seems that the field phi and bar phi are not complex conjugate. If so, this is not stated explicitly, nor is it explained (standard treatments like reference 16 involve complex conjugate fields). Another example is Eq. 35 which needs a more general argument that also includes interactions. Perhaps the most confusing part (even for an occasional practitioner of Keldysh field theory) is the discussion following Eq. 36. The notation \mu_{\nu r} is not explained (although it can be found in the review paper). The transformation from phi^c/q to phi and chi fields is vague and confusing. The end result appears to be more conventional in terms of fields and their complex conjugates, but the reader is left puzzled as to how to get ther. The dots in Eq. 37b are not defined either. An explicit derivation of Eq. 38 too would be helpful to the reader.
3. Scaling dimension, quantum scaling and entanglement: The scaling dimensions of the field phi and chi are derived from the linear part of Eq. 40 and are both given by -d/2. I wonder if this assumes nonzero alpha, given that the phase transition is at alpha=0. Shouldn’t this change the scaling? The authors also make a comment about the connection to “quantum” scaling where the classical and quantum fields have the same scaling dimension. Is there a significance to this observation? Perhaps! The authors basically find that quantum coherences act like a relevant perturbation (to the extent that it changes the universality). If they were irrelevant, one would recover the classical universality class. I am wondering if this could be tied the above quantum scaling. On the same note, there might be interesting features of entanglement. In a generic mixed steady state, quantum coherences are often absent or at least not relevant. The dark-state phase transition considered in this paper seems like a promising avenue to look for nontrivial quantum scaling and entanglement.
4. Can the authors take a limit of Eq. 40 that recovers the classical directed percolation universality? Would it be straightforward to establish a connection to Eq. 23 for example?
5. The authors comment on the emergence of limit cycles and spiral sink and sources. It would be helpful to the reader to show a phase diagram with these phases and indicate the phase transitions them. I am curious if the phase transition to a limit cycle is always first order.

Recommendation

Ask for minor revision

---

## Round 3 · Author Response

Thank you for your review of our manuscript. We also thank the referees for their suggestions for improving the manuscript. We have expanded various sections of our manuscript in response to the referees comments; our replies to specific referee comments and the specific modifications to our manuscript accompanying them are detailed below.
With best regards,
Foster Thompson and Alex Kamenev

---

## Round 3 · List of Changes

Response to Referee 1:
1: The constraints of the action are not known (to us) to be reducible to a symmetry, but are instead imposed by a combination of the dark state FDT-like condition Eq. (35) and the requirement that the stationary state be fluctuationless, which is encoded by Eq. (39). A paragraph following Eq. (40) has been added to clarify this point.
2: The cubic vertices derive from both coherent and incoherent processes $\sim\hat a^\dagger\hat a\hat a$ in which the particle number on one side of the density matrix is changed. In this sense, they can be loosely interpreted as branching and fusion processes on one side of the density matrix at a time. Some comments were added to section 4 in the paragraph following Eq. 41 and the caption of Fig. 6 to this point.
3: The the existence of large-N theories of either the classical or quantum population models is an interesting question which, to our knowledge, has not been thoroughly addressed. It is believed that absorbing state phase transitions in classical population models with multiple species generally fall into the directed percolation universality class (however there may be multi-critical points where the scaling is modified). How this generalizes to the quantum setting and whether or not it admits a large-N description may be interesting questions for future work. We have expanded on this point in our conclusion section.
Response to Referee 2:
1 and 2: We thank the referee for bringing these references to our attention. We expect the nature of our phase transition to be similar to classical directed percolation: the phase transition should persist in all dimensions. Above the critical dimension $d\geq4$ the scaling will become mean-field (but may occur at finite critical ratio due to fluctuations, as shown by the references the referee cites). Higher than cubic vertices may become relevant for $d<3$ which may modify the scaling, however just as in DP the cubic vertices will remain be the most relevant interactions and determine the universality class. A paragraph directly preceding section 4.1 was added to clarify these points.
3: Axes and axis labels have been included in phase portrait plots.
Response to Referee 3:
1: It is a good point that many known Lindbladian theories with both Hamiltonian and incoherent terms display critical behavior in some known classical (equilibrium or non-equilibrium) universality class. The explicit breaking of weak symmetry does not guarantee a new universality class (for one, the a symmetry-breaking perturbation may turn out to be irrelevant, leading to its recovery in the IR; it also may turn out that any new terms permitted by the absence of symmetry do not modify the critical scaling). It nevertheless remains a possibility that the symmetry-breaking perturbation is relevant and drives the theory to a new universality class; we argue that the theory we present is one such example. We agree that, as it is currently written, our discussion of this point does clearly convey this point and thus may be confusing or misleading. The part of the introduction discussing this point has been rewritten to better reflect past results and contextualize our work in relation to them.
2: The discussion in section 3.2 has been expanded and several remarks added throughout to clarify the various points raised. A footnote has been added demonstrating the validity of Eq. (35) for general interactions.
3: The quantum scaling does not assume nonzero $\alpha$. That is, even at the critical point one finds that the scaling dimensions of the classical $\phi$ and quantum $\chi$ fields are equal. The exact value $-d/2$ is only the bare scaling dimension and does receive perturbative corrections at the critical point. A comment has been added to section 4 to emphasize this point.
It is unclear what relation the RG relevance of the quantum processes (coherent or incoherent) to the classical population dynamics has to the quantum scaling. As noted in the text, the Reggeon field theory describing the completely classical directed percolation transition has the same property, so it is not unique to quantum models.
The relationship of dark state transitions to entanglement is an interesting question and we agree that it is something deserving of further study. A paragraph has been added in the conclusion section briefly discussing this point.
4: The classical directed percolation action cannot be retrieved as a limit of the quantum population action. The physical meaning of the fundamental fields is different: the classical field $n$ is the population number while the quantum action is written in terms of the complex coherent state field $\phi\sim\sqrt n$; this point is discussed at the end of section 3.2.
5: Limit cycles in our theory do not occur in full phases, but rather occur at the separation between the active and dark phases. To clarify this, and other details about the phase diagram, we have modified the language at several places in section 4.2 and 4.3 and also included an additional figure with examples of mean-field phase diagrams.

---

## Editorial Decision

published